# How Transformers Solve Propositional Logic Problems: A Mechanistic Analysis

## Abstract

Large language models (LLMs) have shown amazing performance on tasks that require planning and reasoning. Motivated by this, we investigate the internal mechanisms that underpin a network's ability to perform complex logical reasoning. We first construct a synthetic propositional logic problem that serves as a concrete test-bed for network training and evaluation. Crucially, this problem demands nontrivial planning to solve. We perform our study on two fronts. First, we pursue an understanding of precisely how a three-layer transformer, trained from scratch and attains perfect test accuracy, solves this problem. We are able to identify certain "planning" and "reasoning" circuits in the network that necessitate cooperation between the attention blocks to implement the desired logic. Second, we study how a pretrained LLM, Mistral 7B, solves this problem. Using activation patching, we characterize internal components that are critical in solving our logic problem. Overall, our work systemically uncovers novel aspects of small and large transformers, and continues the study of how they plan and reason.

## 1    Introduction

Language models using the transformer architecture (Vaswani et al., 2017) have shown remarkable capabilities on many natural language tasks (Brown et al., 2020; Radford et al., 2019b). Trained with causal language modeling wherein the goal is next-token prediction on huge amounts of text, these models exhibit deep language understanding and generation skills. An essential milestone in the pursuit of models which can achieve a human-like artificial intelligence, is the ability to perform human-like reasoning and planning in complex unseen scenarios. While some recent works using probing analyses have shown that the activations of the deeper layers of a transformer contain rich information about certain mathematical reasoning problems (Ye et al., 2024), the question of what mechanisms inside the model enables such abilities remains unclear.

While the study of how transformers reason in general remains a daunting task, in this work, we aim to improve our *mechanistic* understanding of how a Transformer reason through simple propositional logic problems. For concreteness' sake, consider the following problem:

> Rules: A or B implies C. D implies E. Facts: A is true. B is false. D is true.
> Question: what is the truth value of C?

An answer with *minimal* proof is "A is true. A or B implies C; C is true."

The reasoning problem, while simple-looking on the surface, requires the model to perform several actions that are essential to more complex reasoning problems, all without chain of thought (CoT). Before writing down any token, the model has to first discern the *rule* which is being queried: in this case, it is "A or B implies C". Then, it needs to rely on the premise variables A and B to the locate the relevant *facts*, and find "A is true" and "B is false". Finally, it needs to decide that "A is true" is the correct one to invoke in its answer due to the nature of disjunction. It follows that, to write down the first token "A", the model already has to form a "mental map" of the variable relations, value assignments and query! Therefore, we believe that this is close to the minimal problem to examine how a model internalizes and plans for solving a nontrivial mathematical reasoning problem where apparent ambiguities in the problem specification cannot be resolved trivially.

In this work, we are primarily interested in understanding how transformers solve simple reasoning problems of this form from the perspective of *circuit analysis* (Wang et al., 2023; Rauker et al., 2023): we aim to understand how the transformer utilizes its internal components (attention heads, MLPs, etc.) to execute the causal path of "QUERY→Answer". We perform two flavors of experiments. The first is on shallow transformers trained purely on the synthetic propositional logic problems, where we mainly rely on linear probing and causal interventions to understand the model's reasoning strategy. The other set of experiments are on a pre-trained LLM (Mistral-7B), where we primarily rely on causal intervention techniques and examining the attention statistics to discover and verify the circuits for explaining the reasoning actions of the model.

At a high level, we make the following discoveries based on our two fronts of analysis:

1. (§3) We discover that small transformers, trained purely on the synthetic problem, utilize certain "*routing embeddings*" to significantly alter the information flow of the deeper layers when solving different sub-categories of the reasoning problem. We also characterize the different reasoning pathways: we find that problems querying for reasoning chains involving logical operators typically require greater involvement of all the layers in the model.

2. (§4) Of particular interest is our characterization of the circuit which the pretrained LLM Mistral-7B-v0.1 employs to solve the minimal version of the reasoning problem. We find four families of attention heads which have surprisingly specialized roles in processing different sections of the context: queried-rule locating heads, queried-rule mover heads, fact-processing heads, and decision heads. We find evidence suggesting that the model follows the natural reasoning path of "QUERY→Relevant Rule→Relevant Fact(s)→Decision". We present the circuit *discovery* process in §4.3.1, and circuit *verification* in §4.3.2.

In particular, our analysis of Mistral-7B in §4 is, to our knowledge, the first to characterize the critical components of the circuit employed by an LLM *in the wild* for solving a nontrivial logical reasoning problem (involving distracting clauses) that requires a correct execution of the reasoning path "Query→Relevant rule(s)→Relevant fact(s)→Decision" fully in context. Furthermore, we make concrete progress towards general techniques of verifying the necessity and sufficiency of a (general-purpose) pretrained LLM's reasoning circuit.

Additionally, we define the scope of our analysis as follows. First, in the shallow transformer experiments, we focus on the variant which only has self-attention layers in addition to layer normalization, positional encoding, embedding and softmax parameters. While we could have also included MLP layers, we choose not to because the no-MLP models already achieve 100% *test* accuracy on the problem, and adding MLPs would unnecessarily complicate the analysis. As a second way to focus the scope of paper, in the Mistral-7B experiments, we do *not* seek to uncover *every* model component that participates in solving the reasoning problem. Instead, we focus on those which have the strongest causal influence on the model's *QUERY-sensitive* reasoning actions. By doing so, we can fully justify the necessity of these key components for the model's reasoning actions, without guessing about the roles of less-impactful sub-circuits (for handling low-level text processing, for instance).

## 1.1 RELATED WORKS

**Mechanistic interpretability**. Our work falls in the area of mechanistic interpretability, which aims to understand the mechanisms that enable capabilities of the LLM; such studies involve uncovering certain "circuits" in the network (Elhage et al., 2021; Olsson et al., 2022; Meng et al., 2022; Vig et al., 2020; Feng & Steinhardt, 2024; Wu et al., 2023; Wang et al., 2023; Hanna et al., 2024; Merullo et al., 2024; McGrath et al., 2023; Singh et al., 2024; Feng et al., 2024). While the definition of a "circuit" varies across different works, in this paper, our definition is similar to the one in Wang et al. (2023): it is a collection of model components (attention heads, neurons, etc.) with the "edges" in the circuit indicating the information flow between the components in the forward pass; the "excitation" of the circuit is the input tokens.

**Evaluation of reasoning abilities of LLMs**. Our work is also related to the line of work which focus on empirically evaluating the reasoning abilities of LLMs across different types of tasks (Xue et al., 2024; Chen et al., 2024; Patel et al., 2024; Morishita et al., 2023; Seals & Shalin, 2024; Zhang et al., 2023; Saparov & He, 2023; Saparov et al., 2024; Luo et al., 2024; Han et al., 2024; Tafjord

et al., 2021; Hendrycks et al., 2021; Dziri et al., 2024; Yang et al., 2024). While these studies primarily benchmark their performance on sophisticated tasks, our work focuses on understanding "how" transformers reason on logic problems accessible to fine-grained analysis.

**Analysis of how LLMs reason**. There are far fewer studies that focus on providing fine-grained analysis of *how* LLMs reason. To the best of our knowledge, only a handful of works, such as Xue et al. (2024); Zečević et al. (2023); Ye et al. (2024), share similar goals of understanding how transformers perform multi-step reasoning through detailed empirical or theoretical analysis. However, none studies the [Variable relationships]+[Variable value assignment]+[Query] type problem in conjunction with analysis on *both* small transformers trained purely on the synthetic problem, and large language models trained on a large corpus of internet data.

**Activation patching**. At its core, activation patching, a.k.a. *causal mediation analysis* (Vig et al., 2020; Meng et al., 2022; Hase et al., 2024; Heimersheim & Nanda, 2024; Zhang & Nanda, 2024), uses causal interventions for uncovering the internal mechanisms or "circuits" of LLMs that enable them to perform certain tasks. Typically, the LLM is run on pairs of "original" and "altered" prompts, and we search for components inside the model that "alter" the model's "original behavior" by replacing parts of the model's activation with "altered activations" when running on the original prompts. The opposite "altered→original" intervention can also be adopted.

## 2 DATA MODEL: A PROPOSITIONAL LOGIC PROBLEM

In this section, we describe the synthetic propositional logic problem that shall be the data model of this paper. Our problem follows an implicit causal structure, as illustrated in Figure 1. The structure consists of two distinct chains: exactly one containing a logical operator at the end of the chain, and one forming a purely linear chain. The two chains do *not* share any proposition variable.

We require the model to generate a *minimal* reasoning chain, consisting of "relevant facts", proper rule invocations, and intermediate truth values, to answer the truth-value query. Consider an example constructed from the causal graph in Figure 1, written in English:

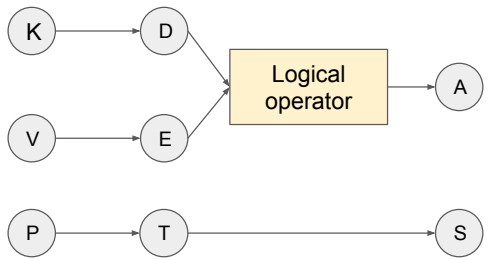

- Rules: K implies D. D or E implies A. V implies E. T implies S. P implies T.

- Facts: K is true. P is true. V is false.

- Query: A.

- Answer: K is true. K implies D; D is true. D or E implies A; A is true.

Figure 1: Synthetic data model. The causal structure has two chains: one with a logical operator (LogOp) at the end and the other being purely a linear causal chain. They do not share any proposition variable. This example is the length-3 case and all symbols are explained in §2.

In this example, the QUERY token A is the terminating node of the OR chain. Since any *true* input to an OR gate (either D or E) results in A being *true*, the minimal solution chooses only one of the starting nodes from the OR chain to construct its argument: in this case, node K is chosen.

**Minimal proof and solution strategy**. In general, the problem requires a careful examination of the rules, facts and query to correctly answer the question. First, the QUERY token determines the chain to deduce its truth value. Second, if it is the logical-operator (LogOp) chain being queried, the model needs to check the facts to determine the correct facts to write down at the start of the reasoning steps (this step can be skipped for queries on the linear chain). Third, the proof requires invoking the rules to properly deduce the truth value of the query token.

**Importance of the first answer token**. Correctly writing down the first answer token is central to the accuracy of the proof, because as discussed in §1, it requires the model to *process every part of the context properly without CoT* due to the minimal-proof requirement of the solution.

## 3 THE REASONING CIRCUIT IN A SMALL TRANSFORMER

In this section, we study how small GPT-2-like transformers, trained solely on the logic problem, approach and solve it. While there are many parts of the answer of the transformer which can lead to interesting observations, in this work, we primarily focus on the following questions:

1. How does the transformer mentally process the context and plan its answer before writing down any token? In particular, *how does it use its "mental notes" to predict the crucial first token*?

2. How does the transformer determine the truth value of the query at the end?

We pay particular attention to the first question, because as noted in §2, *the first answer token reveals the most about how the transformer mentally processes all the context information without any access to chain of thought (CoT)*. We delay the less interesting answer of question 2 to the Appendix due to space limitations.

### 3.1 LEARNER: A DECODER-ONLY ATTENTION-ONLY TRANSFORMER

In this section, we study decoder-only attention-only transformers, closely resembling the form of GPT-2 (Radford et al., 2019a). We train these models exclusively on the synthetic logic problem. The LogOp chain is queried 80% of the time, while the linear chain is queried 20% of the time during training. Details of the model architecture are provided in Appendix B.2.

**Architecture choice for mechanistic analysis**. We select a 3-layer 3-head transformer to initiate our analysis since it is the smallest transformer that can achieve 100% *test* accuracy; we also show the accuracies of several candidate model sizes in Figure 6 in Appendix B for more evidence. Note that a model's answer on a problem is considered accurate only if every token in its answer matches that of the correct answer. Please refer to Appendix B.2 for an illustration of the model components.

### 3.2 MECHANISM ANALYSIS

The model approximately follows the strategy below to predict the first answer token:

1. (Linear vs. LogOp chain) At the QUERY position, the layer-2 attention block sends out a special "routing" signal to the layer-3 attention block, which informs the latter whether the chain being queried is the linear one or not. The third layer then acts accordingly.

2. (Linear chain queried) If QUERY is for the linear chain, the third attention block focuses almost 100% of its attention weights on the QUERY position, that is, it serves a simple "message passing" role: indeed, layer-2 residual stream at QUERY position already has the correct (and linearly decodable) answer in this case.

3. (LogOp chain queried) The third attention block serves a more complex purpose when the LogOp chain is queried. In particular, the first two layers construct a partial answer, followed by the third layer refining it to the correct one.

We illustrate the overall reasoning strategy and core evidence for it in Figure 8 in Appendix B.3.

### 3.2.1 LINEAR OR LOGOP CHAIN: ROUTING SIGNAL AT THE QUERY POSITION

The QUERY token is likely the most important token in the context for the model: it determines whether the linear chain is being queried, and significantly influences the behavior of the third attention block. The transformer makes use of this token in its answer in an intriguing way.

**Routing direction at QUERY**. There exists a "routing" direction $h_{route}$ present in the embedding generated by the layer-2 attention block, satisfying the following properties:

1. $\alpha_1(X)h_{route}$ is present in the embedding when the linear chain is queried, and $\alpha_2(X)h_{route}$ is present when the LogOp chain is queried, where the two $\alpha_i(X)$'s are sample dependent, and satisfy the property that $\alpha_1(X) > 0$, and $\alpha_2(X) < 0$.

2. The "sign" of the $h_{route}$ signal determines the "mode" which layer-3 attention operates in at the ANSWER position. When a sufficiently "positive" $h_{route}$ is present, layer-3 attention acts as if

QUERY is for the linear chain by placing significant attention weight at the QUERY position. A sufficiently "negative" $\boldsymbol{h}_{route}$ causes layer-3 to behave as if the input is the LogOp chain: the model focuses attention on the rules and fact sections, and in fact outputs the correct first token of the LogOp chain!

We discuss our empirical evidence below to support and elaborate on the above mechanism.

*Evidence 1a: chain-type disentanglement at QUERY*. We first observe that, at the QUERY position, the layer-2 attention block's output exhibits disentanglement in its output direction depending on whether the linear or LogOp chain is being queried, as illustrated in Figure 2.

To generate Figure 2, we constructed 200 samples, with the first half querying the linear chain and the second half querying the LogOp chain. We then extracted the layer-2 self-attention block output at the QUERY position for each sample, and calculated the pairwise cosine similarity between these outputs.

*Evidence 1b: distinct layer-3 attention behavior w.r.t. chain type*. When the linear chain is queried, the layer-3 attention heads predominantly focus on the QUERY position, with over 90% of their attention weights on the QUERY position on average (based on 1k test samples). In contrast, when the LogOp chain is queried, less than 5% of layer-3 attention is on the QUERY on average. Instead, attention shifts to the Rules and Facts sections of the context, as shown in Figure 9 in Appendix B.4.

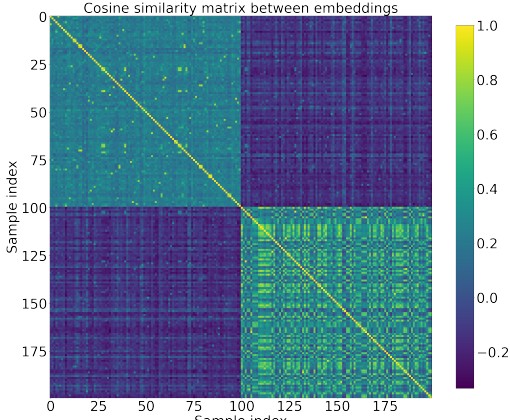

Figure 2: Cosine similarity matrix between output embeddings from layer-2 attention block. Samples 0 to 99 query for the linear chain, samples 100 to 199 query for the LogOp chain. Observe the in-group clustering in angle (top left and bottom right), and the negative cross-group cosine similarity (top right and bottom left).

Observations 1a and 1b suggest that given a chain type (linear or LogOp), certain direction(s) in the layer-2 embedding significantly influences the behavior of the third attention block in the aforementioned manner. We confirm the existence and role of this special direction and reveal more intriguing details below.

*Evidence 1c: computing $\boldsymbol{h}_{route}$, and proving its role with interventions*. To *erase* the instance-dependent information, we *average* the output of the second attention block over 1k samples where QUERY is for the linear chain. We denote this *estimated* average as $\hat{\boldsymbol{h}}_{route}$ which effectively preserves the sample-invariant signal. To test the influence of $\hat{\boldsymbol{h}}_{route}$, we investigate its impact on the model's reasoning process, and we observe two intriguing properties:

1. (Linear→LogOp intervention) We generate 500 test samples where QUERY is for the linear chain. *Subtracting* the embedding $\hat{\boldsymbol{h}}_{route}$ from the second attention block's output causes the model to consistently predict the correct first token for the *LogOp chain* on the test samples. In other words, the "mode" in which the model reasons is flipped from "linear" to "LogOp".

2. (LogOp→linear intervention) We generate 500 test samples where QUERY is for the LogOp chain. *Adding* $\hat{\boldsymbol{h}}_{route}$ to the second attention block's output causes the three attention heads in layer 3 to focus on the QUERY position: greater than 95% of the attention weights are on this position averaged over the test samples. In this case, however, the model does not output the correct starting node for the linear chain on more than 90% of the test samples.

It follows that $\boldsymbol{h}_{route}$ indeed exists, and the "sign" of it determines the attention patterns in layer 3 (and the overall network's output!) in the aforementioned manner.

### 3.2.2 ANSWER FOR THE LINEAR AND LOGOP CHAIN

**Linear chain: answer at layer-2 residual stream at QUERY position**. At this point, it is clear to us that, *when QUERY is for the linear chain*, the third layer mainly serves a simple "message

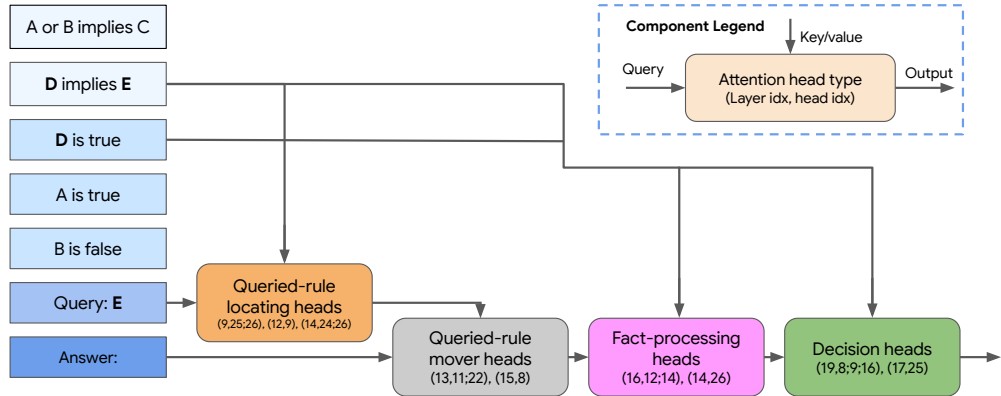

Figure 3: High-level properties of Mistral-7B's reasoning circuit. The (chunks of) input tokens are on the left, which are passed into the residual stream and processed by the attention heads. We illustrate the information flow manipulated by the different types of attention heads we identified to be vital to the reasoning task.

passing" role: it passes the information in the layer-2 residual stream at the QUERY position to the ANSWER position. One natural question arises: does the input to the third layer truly contain the information to determine the first token of the answer, namely the starting node of the linear chain? The answer is yes.

*Evidence 2: linearly-decodable linear-chain answer at layer 2*. We train an affine classifier with the same input as the third attention block at the QUERY position, with the target being the start of the linear chain; the training samples only query for the linear chain, and we generate 5k of them. We obtain a test accuracy above 97% for this classifier (on 5k test samples), confirming that layer 2 already has the answer linearly encoded at the QUERY position. To add further contrasting evidence, we train another linear classifier with exactly the same task as before, except it needs to predict the correct start of the *LogOp* chain. We find that the classifier achieves a low test accuracy of approximately 27%, and exhibits severe overfitting with the training accuracy around 94%.

**LogOp chain: partial answer in layers 1 & 2 + refinement in layer 3**. To predict the correct starting node of the LogOp chain, the model employs the following strategy:

1. The first two layers encode the LogOp and only a "partial answer". More specifically, we find evidence that (1) when the LogOp is an AND gate, layers 1 and 2 tend to pass the node(s) with FALSE assignment to layer 3, (2) when the LogOp is an OR gate, layers 1 and 2 tend to pass node(s) with TRUE assignment to layer 3.

2. The third layer, combining information of the two starting nodes of the LogOp chain, and the information in the layer-2 residual stream at the ANSWER position, output the correct answer.

We delay the full set of evidence for the above two claims to Appendix B.4 due to space limits.

## 4   THE REASONING CIRCUIT IN MISTRAL-7B

We now turn to examine how a pretrained LLM, namely Mistral-7B-v0.1, solves this reasoning problem. We choose this LLM as it is amongst the smallest accessible model which achieves above 70% accuracy on (a minimal version of) our problem. Our primary focus here is the same as in the previous section: how does the model infer the first answer token without any CoT? We are interested in this question as the first answer token requires to model to process all the information in the context properly without access to any CoT.

We illustrate the reasoning circuit inside the model for this prediction task in Figure 3. At a high level, there are two intriguing properties of the reasoning circuit of the LLM:[1]

---

[1]We use $(\ell, h)$ to denote an attention head. When referencing multiple heads in the same layer, we write $(\ell, h_1; h_2; ...; h_n)$ for brevity.

1. Compared to the attention blocks, the MLPs are relatively unimportant to correct prediction.

2. There is a sparse set of attention heads that are found to be central to the reasoning circuit: the queried-rule locating heads, queried-rule mover heads, fact-processing heads, and decision heads. We discuss circuit discovery in §4.3.1, and circuit verification in §4.3.2.

### 4.1 MINIMAL PROBLEM DESCRIPTION

In our Mistral-7B experiments, the input samples have the following properties:

1. We give the model 6 (randomly chosen) in-context examples before asking for the answer.

2. The problem is length-2: only one rule involving the OR gate, and one linear-chain rule. Moreover, the answer is always true. In particular, the truth values of the two premise nodes of the OR chain always have one FALSE and one TRUE.

3. The proposition variables are all (single-token) capital English letters.

The design decision in the first point is to ensure fairness to the LLM which was not trained on our specific logic problem. As for the last two points, we restrict the problem in this fashion mainly to ensure that the first answer token is *unique*, which improves the tractability of the analysis. Note that these restrictions do not take away the core challenge of this problem: *the LLM still needs to process all the context information without CoT to determine the correct first token.* We discuss concrete examples and finer details of testing the model on this problem in Appendix C.

### 4.2 CAUSAL MEDIATION ANALYSIS

We provide evidence in this part of the paper primarily relying on a popular technique in mechanistic interpretability: *causal mediation analysis*. Our methodology is roughly as follows:

1. Suppose we are interested in the role of the activations of certain components of the LLM in a certain (sub-)task. For a running example, say we want to understand what role the attention heads play in processing and passing QUERY information to the ":" position for inference. Let us denote the activations as $\boldsymbol{A}_{\ell,h;t}(\boldsymbol{X})$, representing the activation of head $h$ in layer $\ell$, at token position $t$.

2. Typically, the analysis begins by constructing two sets of prompts which differ in subtle ways. A natural construction in our example is as follows: define sets of samples $\mathcal{D}_{orig}$ and $\mathcal{D}_{alt}$, where $\boldsymbol{X}_{orig,n}$ and $\boldsymbol{X}_{alt,n}$ have exactly the same context, except in $\boldsymbol{X}_{orig,n}$, QUERY is for the LogOp chain, while in $\boldsymbol{X}_{alt,n}$, QUERY is for the linear chain. Moreover, denote the correct targets $y_{orig,n}$ and $y_{alt,n}$ respectively.

3. We run the LLM on $\mathcal{D}_{orig}$ and $\mathcal{D}_{alt}$, caching the attention-head activations. We also obtain the logits of the model. We can compute the model's *logit differences*

$$\begin{aligned}
\Delta_{orig,n} &= \text{logit}(\boldsymbol{X}_{orig,n})[y_{orig,n}] - \text{logit}(\boldsymbol{X}_{orig,n})[y_{alt,n}], \\
\Delta_{alt,n} &= \text{logit}(\boldsymbol{X}_{alt,n})[y_{alt,n}] - \text{logit}(\boldsymbol{X}_{alt,n})[y_{orig,n}].
\end{aligned} \tag{1}$$

For a high-accuracy model, $\Delta_{orig,n}$ and $\Delta_{alt,n}$ should be *large and positive* for most $n$'s, since most of the time it must be able to clearly tell that on an $\boldsymbol{X}_{orig,n}$, it is the LogOp chain which is being queried, not the linear chain (similarly on an $\boldsymbol{X}_{alt,n}$).

4. We now perform intervention for all $n, \ell, h$ and $t$:

   (a) Run the model on $\boldsymbol{X}_{orig,n}$, however, replacing the original activation $\boldsymbol{A}_{\ell,h;t}(\boldsymbol{X}_{orig,n})$ by the altered $\boldsymbol{A}_{\ell,h;t}(\boldsymbol{X}_{alt,n})$. Now let the rest of the run continue.[2] Let us denote the logits obtained in this intervened run as $\text{logit}^{\to alt;(\ell,h,t)}(\boldsymbol{X}_{orig,n})$.

   (b) Now compute the intervened logit difference

$$\Delta_{orig\to alt,n;(\ell,h,t)} = \text{logit}^{\to alt;(\ell,h,t)}(\boldsymbol{X}_{orig,n})[y_{alt,n}] - \text{logit}^{\to alt;(\ell,h,t)}(\boldsymbol{X}_{orig,n})[y_{orig,n}].$$

5. Average the $\Delta_{orig\to alt,n;(\ell,h,t)}$'s over $n$ for every $\ell, h$ and $t$ (recall that $n$ is the sample index).

---

[2] Please note that layers $\ell + 1$ to $L$ are influenced at and after token position $t$, and technically speaking, now operate "out of distribution".

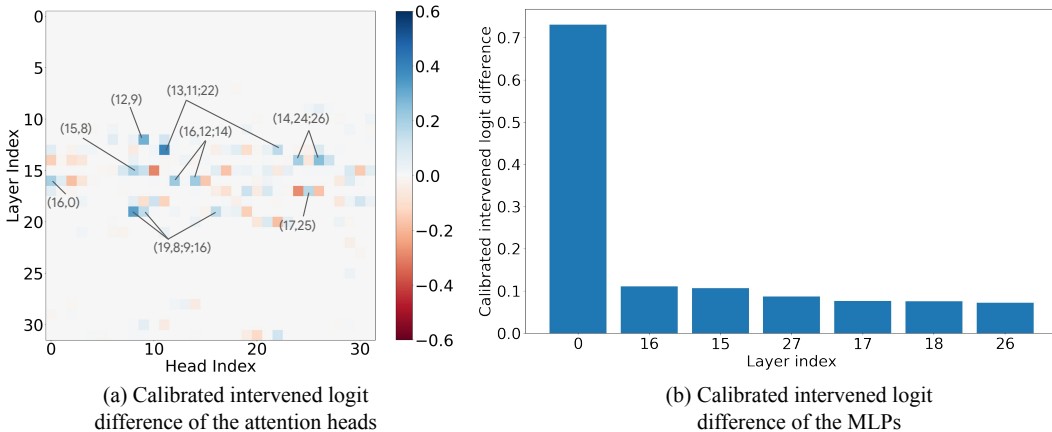

(a) Calibrated intervened logit difference of the attention heads

(b) Calibrated intervened logit difference of the MLPs

Figure 4: Attention head and MLP output patching results (over all token positions in relevant context) in (a) and (b) respectively. The closer to 1, the greater the causal influence a component has on the LLM's correct inference. We highlight the heads with the highest intervened logit difference in (a). We observe that only a small set of attention heads, and the first MLP layer exhibit strong causal influence on the model's correct inference.

6. This procedure helps us identify components that are significant in processing the QUERY information for inference. Intuitively, an activation that result in a positive and large $\Delta_{orig \to alt, n;(\ell, t)}$ (with an ideal upper limit being $\Delta_{alt, n}$) play a significant role in this subtask, because this activation helps "altering" the model's "belief" from "QUERY is for the LogOp chain" to "QUERY is for the linear chain". **We denote the components identified as a set $\mathcal{C}$.**

7. *Remark*: due to the symmetry of this running example, it is perfectly sensible to perform $alt \to orig$ interventions too, by mirroring the above procedures; we indeed adopt this mirrored procedure too in our experiments for efficient use of model computations.

**QUERY-based activation patching is important to reasoning circuit discovery and verification**. To justify this claim, there are two points to emphasize first. (1) To solve the reasoning problem, the QUERY is critical to *initiating* the reasoning chain: without it, the rules and facts are completely useless; with it, the reasoner can then proceed to identify the relevant rules and facts to predict the answer. (2) The prompt pairs differ *only* by the QUERY token: there is complete information for the model to provide answer for both chains given the QUERY.

If performing the aforementioned interventions on a model component leads to a large intervened logit difference (i.e. it alters the model's "belief"), then this component must be integral to the reasoning circuit, because the component is now identified to be QUERY-sensitive *and* has causal influence on (parts of) the model's reasoning actions. Furthermore, if on (most of) the prompt pairs, by patching all the circuit components in $\mathcal{C}$ simultaneously while freezing the rest causes the intervened logit difference to approach the "maximal" altered logit difference[3], then we obtain evidence suggesting *sufficiency* of $\mathcal{C}$ for explaining the LLM's (QUERY-sensitive) reasoning actions.

For greater clarity, we provide illustrations of the intervention experiments in Appendix C.2.

## 4.3 CIRCUIT ANALYSIS

In this section, we discuss properties of the reasoning circuit of Mistral-7B. The order by which we present the results will roughly follow the process which we discovered and verified the circuit; we believe this adds greater transparency to the circuit analysis process. We delay the more involved (and complete) set of experimental results to Appendix C.

### 4.3.1 CIRCUIT DISCOVERY: NECESSITY-BASED SEARCH FOR MODEL COMPONENTS

We initiate our analysis with QUERY-based patching, following the same procedure as detailed in §4.2. In this set of experiments, we discover the main attention heads responsible for processing the context and performing inference as introduced in the beginning of this Section.

---

[3]In this case, we simply consider the "maximal" altered logit difference to be the logit difference of the model run on the *altered* prompts without interventions, i.e. the $\Delta_{alt, n}$'s.

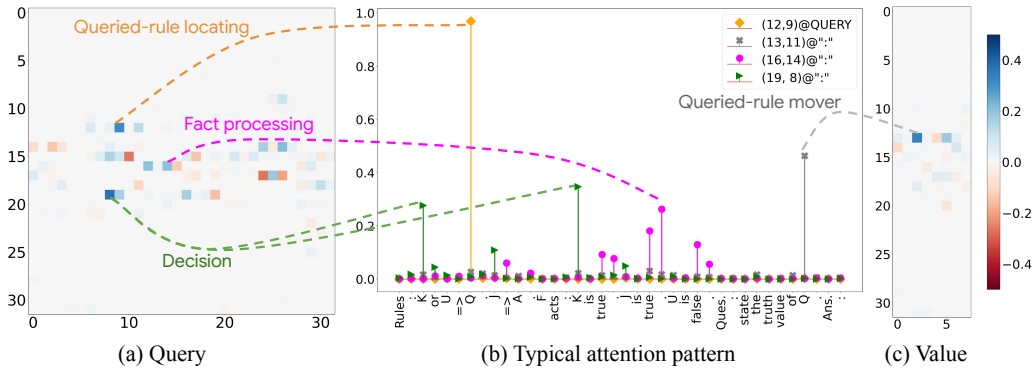

(a) Query  (b) Typical attention pattern  (c) Value

Figure 5: Patching of query and value activations in (a) and (c); we found that intervening the key activations only yield trivial scores, so we do not report them here. We show in (b) the typical attention patterns of a representative set of the attention heads which are identified to be important in the intervention experiments shown in (a) and (c). There are several distinct observations which can be made in (b). Queried-rule locating head (12,9): observe that it correctly locates the queried rule which ends with **Q**. Queried-rule mover head (13,11): the only token position which it focuses on is the QUERY token **Q**. Fact processing head (16,14): attention concentrates in the fact section. Decision head (19,8): attention focused on the correct first answer token **K**.

**High-level interventions**. Figure 4(a) helps us locate a small set of attention heads which are central to the "belief altering" of the LLM. More specifically, only attention heads (12,9), (13,11;22), (14,24;26), (16,0;12;14), (17,25), (19,8;9;16), and (9,25;26)[4] are observed with relatively high intervened logit differences. As for the MLPs, shown in Figure 4(b), play little role in this circuit, except for MLP-0. However, MLP-0 had been observed to act more as a "nonlinear token embedding" than a complex high-level processing unit (Wang et al., 2023). In the rest of this section, we primarily devote our analysis to the attention heads, and leave the exact role of the MLPs to future work.

*Remark*. In Figure 4 and the rest, unless otherwise specified, we adopt a calibrated version of the logit difference (see Appendix C): the closer to 1, the more significant the component is in "altering" the "belief" of the model on the selected subtask.

**Attention-head sub-component patching (QUERY-based patching).** We now aim to understand why the attention heads identified in the last sub-section are important. For now, we continue with QUERY altering in the prompt pairs. Through intervening on the sub-components of each attention head, namely their value, key, and query, and through examining details of their attention weights, we find that there are roughly four types of attention heads. We show the results in Figure 5.

1. Queried-rule locating head. Attention head (12,9)'s *query* activation has a large intervened logit difference according to Figure 5(a), therefore, its query and attention patterns are QUERY-dependent and contribute to altering the model's "belief". Furthermore, at the QUERY position, we find that *on average*, its attention weight is above 90% at the "conclusion" variable of the rule being queried. In other words, it is responsible for *locating* the queried rule, and storing that rule's information at the QUERY position.[5]

2. Queried-rule mover head. Attention head (13,11)'s *value* activations have large intervened logit difference, and intriguingly, its query and key activations do *not* share that tendency. This already suggests that its attention pattern performs a fixed action on both the original and altered prompts, and only the value information is sensitive to QUERY. Furthermore, within the relevant context (excluding the 6 in-context examples given), (13,11) assigns above 50% attention weight to the QUERY position, and its attention weight at QUERY is about 10 times larger than the second largest one on average. Recalling the role of layer 12, we find evidence that (13,11) moves the QUERY and queried-rule information to the ":" position.[6]

---

[4]We discover these two heads with a slightly different patching experiment, presented in Appendix C.3.2.

[5](9,25;26), (14,24;26) exhibit similar tendencies, albeit with smaller intervened logit differences.

[6](13,22), (15,8), (16,0) also appear to belong to this type, albeit with smaller intervened logit difference.

3. **Fact processing heads.** Attention heads (16,12;14) and (14,26)'s *query* activations have large intervened logit differences. Within the relevant context, at the ":" token position, their attention weights are above 56, 80 and 63% respectively in the fact section of the context (starting from "Fact" and ending on "." before "Question").

4. **Decision head.** Attention head (19,8)'s query activations have large intervened logit differences. Its attention pattern suggests that it is a "decision" head: within the relevant context, when the model is *correct*, the head's top-2 attention weights are always on the correct starting node of the queried rule and the correct variable in the fact section, and the two token positions occupy more than 60% of its total attention in the relevant context on average. In other words, it already has the answer.[7]

Due to the space limits, we delay detailed inspection and visualization of the attention statistics to Appendix C.3.3, and finer-grained activation patching experiments examining the function of the attention head families to Appendix C.5 (queried-rule locating heads) and C.6 (fact-processing heads and decision heads). Moreover, we discuss some preliminary evidence which shows the surprising similarities of the attention circuit in Gemma-2-9B to Mistral-7B's for solving the logic problem in Appendix C.7.

### 4.3.2 CIRCUIT VERIFICATION: A SUFFICIENCY TEST

A natural question now arises: is $\mathcal{C}$ *sufficient* to explain the (QUERY-sensitive) reasoning actions of the LLM? As explained in §4.2, we quantify sufficiency by measuring the "belief" altering effect of the circuit, and test $\mathcal{C}$ as follows. Given the original and altered prompt pairs $\{(\boldsymbol{X}_{orig,n}, \boldsymbol{X}_{alt,n})\}_{n \in [N]}$ which differ only by QUERY, we aim to verify that, simultaneously patching $\boldsymbol{A}_{\ell,h;t}(\boldsymbol{X}_{orig,n}) \rightarrow \boldsymbol{A}_{\ell,h;t}(\boldsymbol{X}_{alt,n})$ for every attention head in the circuit $\mathcal{C} = \{(12,9),(13,11),(16,12),(19,8),...\}$ at their functioning token positions, while freezing all the other attention heads to the original activations $\boldsymbol{A}_{\ell,h;t}(\boldsymbol{X}_{orig,n})$, lead to the average *circuit-intervened* logit difference $\Delta_{orig \rightarrow alt}^{\mathcal{C}} = \frac{1}{N}\sum_{n=1}^{N}\Delta_{orig \rightarrow alt,n}^{\mathcal{C}}$ approaching or surpassing the average logit difference of the (un-intervened) model run on the altered prompts, namely $\Delta_{alt} = \frac{1}{N}\sum_{n=1}^{N}\Delta_{alt,n}$. We confirm this hypothesis below:

| $\mathcal{C}^{\dagger}$ | $\mathcal{C}_{null}$ | $\mathcal{C}$ | $\mathcal{C} - QRLH$ | $\mathcal{C} - QRMH$ | $\mathcal{C} - FPH$ | $\mathcal{C} - DH$ |
|---|---|---|---|---|---|---|
| $\Delta_{orig \rightarrow alt}^{\mathcal{C}^{\dagger}}/\Delta_{alt}$ | -1.0 | **0.98** | -1.02 | -0.99 | -0.25 | -0.89 |

Table 1: $\Delta_{orig \rightarrow alt}^{\mathcal{C}^{\dagger}}/\Delta_{alt}$, with different choices of $\mathcal{C}^{\dagger}$. $\mathcal{C}_{null}$ denotes the *empty* circuit, i.e. the case where no intervention is performed. We abbreviate the attention head families, for example, $DH =$ decision heads; $\mathcal{C} - DH =$ full circuit but with the decision heads removed.

We find that by patching all 14 attention heads in $\mathcal{C}$, $\Delta_{orig \rightarrow alt}^{\mathcal{C}}$ is about 98% of the "maximal" average logit difference $\Delta_{alt}$ on the altered samples. Moreover, removing any one of the four families of attention heads from $\mathcal{C}$ in the circuit interventions renders the "belief altering" effect of the intervention almost trivial. We present further discussions of the experimental procedure and results, and caveats of reasoning circuit verification in Appendix C.4.

## 5 CONCLUSION

We studied the reasoning mechanisms of both small transformers and LLMs on a synthetic propositional logic problem. We analyzed a shallow decoder-only attention-only transformer trained purely on this problem as well as a pretrained Mistral-7B LLM. We uncovered interesting mechanisms the small and large transformers adopt to solve the problem. For the small models, we found the existence of "routing" signals that significantly alter the model's reasoning pathway depending on the sub-category of the problem instance. For Mistral-7B, we characterized the circuit formed by four families of attention heads that implement the reasoning pathway of "QUERY→Relevant Rule→Relevant Facts→Decision", and make concrete progress towards evaluating the necessity and sufficiency of the reasoning circuit through carefully designed causal interventions. These findings provide valuable insights into the inner workings of LLMs on mathematical reasoning problems.

---

[7](17,25), (19,9;16) exhibit similar tendencies, albeit with smaller intervened logit differences.

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

# Appendix

## Contents

## A  PROPOSITIONAL LOGIC PROBLEM AND EXAMPLES

In this section, we provide a more detailed description of the propositional logic problem we study in this paper, and list representative examples of the problem.

At its core, the propositional logic problem requires the reasoner to (1) distinguish which chain type is being queried (LogOp or linear), and (2) if it is the LogOp chain being queried, the reasoner must know what truth value the logic operator outputs based on the two input truth values.

Below we provide a comprehensive list of representative examples of our logic problem at length 2 (i.e. each chain is formed by one rule). We use [Truth values] to denote the relevant input truth value assignments (i.e. relevant facts) to the chain being queried below.

1. Linear chain queried, [True]
   - Rules: A or B implies C. **D implies E**.
   - Facts: A is true. B is true. **D is true**.
   - Question: what is the truth value of C?
   - Answer: D true. D implies E; E True.

2. Linear chain queried, [False]
   - Rules: A or B implies C. **D implies E**.
   - Facts: A is true. B is true. **D is false**.
   - Question: what is the truth value of C?
   - Answer: D false. D implies E; E undetermined.

3. LogOp chain queried, LogOp = OR, [True, True]
   - Rules: A **or** B implies C. D implies E.
   - Facts: **A is true. B is true.** D is true.
   - Question: what is the truth value of C?
   - Answer: B true. A or B implies C; C True.
   *Remark.* In this case, the answer "A true. A or B implies C; C True" is also correct.

4. LogOp chain queried, LogOp = OR, [True, False]
   - Rules: A **or** B implies C. D implies E.
   - Facts: **A is true. B is false.** D is true.
   - Question: what is the truth value of C?
   - Answer: A true. A or B imples C; C True.

5. LogOp chain queried, LogOp = OR, [False, False]
   - Rules: A **or** B implies C. D implies E.
   - Facts: **A is false. B is false.** D is true.
   - Question: what is the truth value of C?
   - Answer: A false B false. A or B implies C; C undetermined.

6. LogOp chain queried, LogOp = AND, [True, True]
   - Rules: A **and** B implies C. D implies E.
   - Facts: **A is true. B is true.** D is true.
   - Question: what is the truth value of C?
   - Answer: A true B true. A and B implies C; C True.

7. LogOp chain queried, LogOp = AND, [True, False]
   - Rules: A **and** B implies C. D implies E.
   - Facts: **A is true. B is false.** D is true.
   - Question: what is the truth value of C?
   - Answer: B false. A and B implies C; C undetermined.

8. LogOp chain queried, LogOp = AND, [False, False]
   - Rules: A **and** B implies C. D implies E.

- Facts: **A is false. B is false.** D is true.
- Question: what is the truth value of C?
- Answer: A false. A and B implies C; C undetermined.

*Remark.* In this case, the answer "B false. A and B implies C; C undetermined" is also correct.

The length-3 case is a simple generalization of this set of examples, so we do not cover those examples here.

## B  LENGTH-3 SMALL TRANSFORMER STUDY: EXPERIMENTAL DETAILS

### B.1  DATA DEFINITION AND EXAMPLES

As illustrated in Figure 1, the propositional logic problem always involve one logical-operator (LogOp) chain and one linear chain. In this paper, we study the length-3 case for the small-transformer setting, and length-2 case for the Mistral-7B-v0.1 case.

The input context has the following form:

```
RULES_START K implies D. V implies E. D or E implies A.
P implies T. T implies S. RULES_END
FACTS_START K TRUE. V FALSE. P TRUE. FACTS_END
QUERY_START A. QUERY_END
ANSWER
```

and the answer is written as

```
K TRUE. K implies D; D TRUE. D or E implies A; A TRUE.
```

In terms the the English-to-token mapping, RULES_START, RULES_END, FACTS_START, FACTS_END, QUERY_START, QUERY_END ANSWER, . and ; are all unique single tokens. The logical operators and and or and the connective implies are unique single tokens. The proposition variables are also unique single tokens.

*Remark.* The rules and facts are presented in a *random* order in the respective sections of the context in all of our experiments unless otherwise specified. This prevents the model from adopting position-based shortcuts in solving the problem.

Additionally, for more clarity, it is entirely possible to run into the scenario where the LogOp chain is queried, LogOp = OR and the two relevant facts both have FALSE truth values (or LogOp = AND and both relevant facts are TRUE), in which case the answer is not unique. For instance, if in the above example, both K and V are assigned FALSE, then both answers below are logically correct:

```
K FALSE V FALSE. K implies D; D UNDETERMINED. V implies E;
E UNDETERMINED. D or E implies A; A UNDETERMINED.
```

and

```
V FALSE K FALSE.  V implies E; E UNDETERMINED.
K implies D; D UNDETERMINED. D or E implies A; A UNDETERMINED.
```

**Problem specification**. In each logic problem instance, the proposition variables are randomly sampled from a pool of 80 variables (tokens). The truth values in the fact section are also randomly chosen. In the training set, the linear chain is queried 20% of the time; the LogOp chain is queried 80% of the time. We train every model on 2 million samples.

**Architecture choice**. Figure 6 indicates the reasoning accuracies of several candidate model variants. We observe that the 3-layer 3-head variant is the smallest model which achieves 100% accuracy. We found that 3-layer 2-head models, trained of some random seeds, do converge and obtain near 100% in accuracy (typically above 97%), however, they sometimes *fail* to converge. The 3-layer 3-head variants we trained (3 random seeds) all converged successfully.

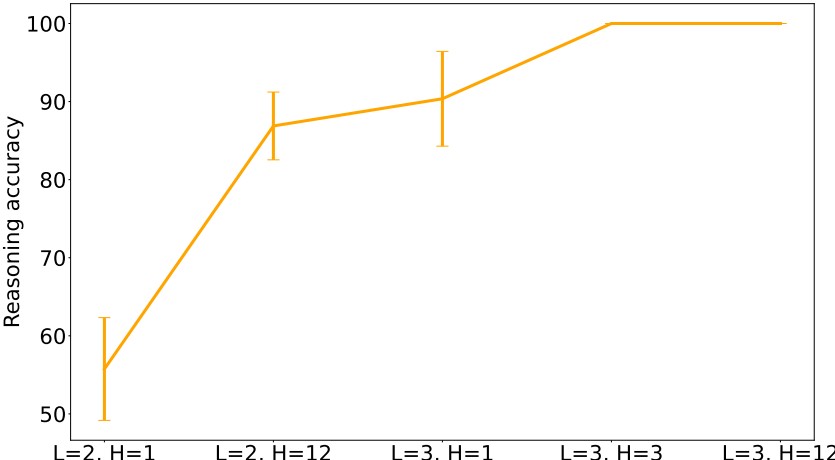

Figure 6: Reasoning accuracies of several models on the length-3 problem. x-axis: model architecture (number of layers, number of heads); y-axis: reasoning accuracy. Note that the 3-layer 3-head variant is the smallest which obtains 100% accuracy on the logic problems.

## B.2 SMALL-TRANSFORMER CHARACTERISTICS, AND TRAINING DETAILS

### B.2.1 TRANSFORMER DEFINITION

The architecture definition follows that of GPT-2 closely. We illustrate the main components of this model in Figure 7, and point out where the frequently used terms in the main text of our paper are in this model.

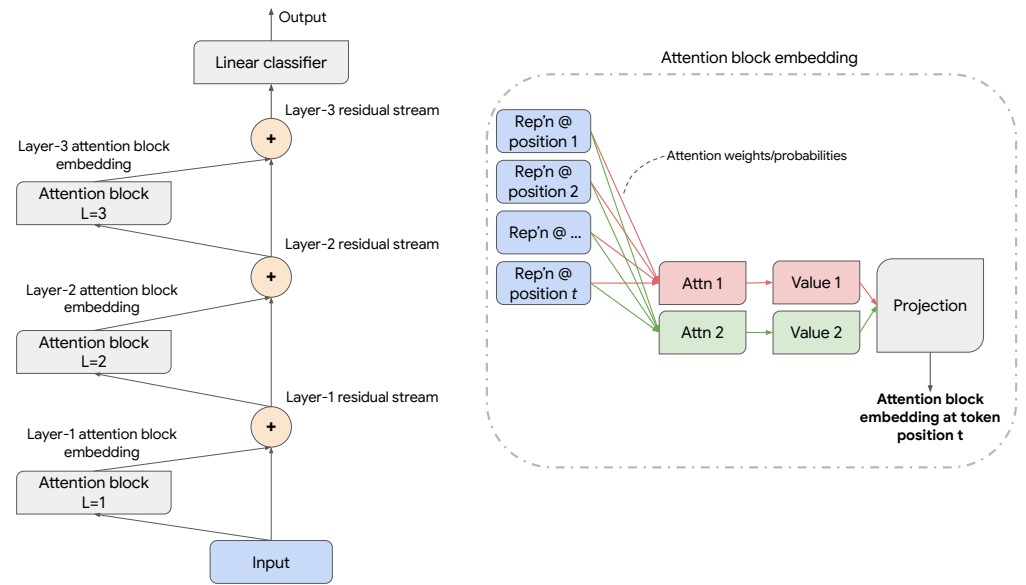

Figure 7: Illustration of the major components of a 3-layer attention-only decoder-only transformer on the left, and a rough "sketch" of what is computed inside an attention block (2 attention heads for simplicity of the sketch).

The following is the more technical definition of the model. Define input $\boldsymbol{x} = (x_1, x_2, ..., x_t) \in \mathbb{N}^t$, a sequence of tokens with length $t$. It is converted into a sequence of (trainable) token embeddings $\boldsymbol{X}_{token} = (\boldsymbol{e}(x_1), \boldsymbol{e}(x_2), ..., \boldsymbol{e}(x_t))^T \in \mathbb{R}^{t \times d_e}$, where we denote the hidden embedding dimension of the model with $d_e$. Adding to it the (trainable) positional embeddings $\boldsymbol{P} = (\boldsymbol{p}_1, \boldsymbol{p}_2, ..., \boldsymbol{p}_t)^T \in$

$\mathbb{R}^{t \times d_e}$, we form the zero-th layer embedding of the transformer

$$\boldsymbol{X}_0 = \boldsymbol{X}_{token} + \boldsymbol{P} = (\boldsymbol{e}(x_1) + \boldsymbol{p}_1, ..., \boldsymbol{e}(x_t) + \boldsymbol{p}_t). \tag{2}$$

This zero-th layer embedding is then processed by the attention blocks as follows.

Let the model have $L$ layers and $H$ heads. For layer index $\ell \in [L]$ and head index $j \in [H]$, attention head $\mathcal{A}_{\ell,j}$ is computed by

$$\mathcal{A}_{\ell,j}(\boldsymbol{X}_{\ell-1}) = \mathcal{S}\left(\text{causal}\left[\frac{1}{\sqrt{d_h}}\left(\boldsymbol{Q}_{\ell,j}\widetilde{\boldsymbol{X}}_{\ell-1}^T\right)^T \boldsymbol{K}_{\ell,j}\widetilde{\boldsymbol{X}}_{\ell-1}^T\right]\right)\widetilde{\boldsymbol{X}}_{\ell-1}\boldsymbol{V}_{\ell,j}^T \in \mathbb{R}^{t \times d_h}, \tag{3}$$

where $d_h = \frac{d_e}{H}$.

We explain how the individual components are computed below.

- Let us begin with how the $\mathcal{S}(...)$ term is computed.

- $\widetilde{\boldsymbol{X}}_{\ell-1} = \text{LayerNorm}(\boldsymbol{X}_{\ell-1}) \in \mathbb{R}^{t \times d_e}$, where LayerNorm denotes the layer normalization operator (Ba et al., 2016).

- $\boldsymbol{Q}_{\ell,j}, \boldsymbol{K}_{\ell,j} \in \mathbb{R}^{d_h \times d_e}$ are the key and query matrices of attention head $(\ell, j)$, where $d_h = \frac{d_e}{H}$. They are multiplied with the input $\widetilde{\boldsymbol{X}}_{\ell-1}$ to obtain the query and key activations $\boldsymbol{Q}_{\ell,j}\widetilde{\boldsymbol{X}}_{\ell-1}^T$ and $\boldsymbol{K}_{\ell,j}\widetilde{\boldsymbol{X}}_{\ell-1}^T$, both in the space $\mathbb{R}^{d_h \times t}$. We then perform the "scaled dot-product" of the query and key activations to obtain

$$\frac{1}{\sqrt{d_h}}\left(\boldsymbol{Q}_{\ell,j}\widetilde{\boldsymbol{X}}_{\ell-1}^T\right)^T \boldsymbol{K}_{\ell,j}, \widetilde{\boldsymbol{X}}_{\ell-1}^T \in \mathbb{R}^{t \times t}, \tag{4}$$

which was introduced in Vaswani et al. (2017) and also used in GPT2 (Radford et al., 2019b).

- The causal mask operator causal : $\mathbb{R}^{t \times t} \to \mathbb{R}^{t \times t}$ allows the lower triangular portion of the input (including the diagonal entries) to pass through unchanged, and sets the upper triangular portion of the input to $-U$, where $U$ is a very large positive number (some papers simply denote this $-U$ as $-\infty$). In other words, given any $\boldsymbol{M} \in \mathbb{R}^{t \times t}$ and $(i, k) \in [t] \times [t]$,

$$\begin{aligned}[\text{causal}\,[\boldsymbol{M}]]_{i,k} &= [\boldsymbol{M}]_{i,k}, \text{ if } i \geq k; \\ [\text{causal}\,[\boldsymbol{M}]]_{i,k} &= -U, \text{ if } i < k.\end{aligned} \tag{5}$$

- $\mathcal{S} : \mathbb{R}^{t \times t} \to \mathbb{R}^{t \times t}$ is the softmax operator, which computes the row-wise softmax output from the input square matrix. In particular, given a square input matrix $\boldsymbol{M} \in \mathbb{R}^{t \times t}$ with its upper triangular portion set to $-U$ (note that the causal mask operator indeed causes the input to $\mathcal{S}$ to have this property), we have

$$\begin{aligned}[\mathcal{S}(\boldsymbol{M})]_{i,k} &= \frac{\exp\left([\boldsymbol{M}]_{i,k}\right)}{\sum_{n=1}^{i}\exp([\boldsymbol{M}]_{i,n})}, \text{ if } i \geq k; \\ [\mathcal{S}(\boldsymbol{M})]_{i,k} &= 0, \text{ if } i < k.\end{aligned} \tag{6}$$

- To recap a bit, we have now explained how to compute the first major term in equation 3, namely $\mathcal{S}\left(\text{causal}\left[\frac{1}{\sqrt{d_h}}\left(\boldsymbol{Q}_{\ell,j}\widetilde{\boldsymbol{X}}_{\ell-1}^T\right)^T \boldsymbol{K}_{\ell,j}\widetilde{\boldsymbol{X}}_{\ell-1}^T\right]\right) \in [0,1]^{t \times t}$. It reflects the attention pattern (also called attention probabilities) of the attention head $(\ell, j)$ illustrated in Figure 7's right half. Intuitively speaking, the $(i, k)$ entry of this $t$ by $t$ matrix reflects how much the attention head moves the information from the previous layer $\ell - 1$ at the source token position of $k$ to the current layer $\ell$ at the target token position $i$.

- Now what about $\widetilde{\boldsymbol{X}}_{\ell-1}\boldsymbol{V}_{\ell,j}^T$? $\boldsymbol{V}_{\ell,j} \in \mathbb{R}^{d_h \times d_e}$ is the value matrix of attention head $(\ell, j)$. It is multiplied with $\widetilde{\boldsymbol{X}}_{\ell-1}$ to obtain the value activation $\widetilde{\boldsymbol{X}}_{\ell-1}\boldsymbol{V}_{\ell,j}^T \in \mathbb{R}^{t \times d_h}$.

- At this point, we have shown how the whole term in equation equation 3 is computed.

Having computed the output of the all $H$ attention heads in the attention block at layer $\ell$, we find the output of the attention block as follows:

$$\boldsymbol{X}_\ell = \boldsymbol{X}_{\ell-1} + \text{Concat}[\mathcal{A}_{\ell,1}(\boldsymbol{X}_{\ell-1}), ..., \mathcal{A}_{\ell,H}(\boldsymbol{X}_{\ell-1})]\boldsymbol{W}_{O,\ell}^T. \tag{7}$$

The operators are defined as follows:

- Concat[·] is the concatenation operator, where $\text{Concat}[\mathcal{A}_{\ell,1}(\boldsymbol{X}_{\ell-1}), ..., \mathcal{A}_{\ell,H}(\boldsymbol{X}_{\ell-1})] \in \mathbb{R}^{t \times d_e}$.

- $\boldsymbol{W}_{O,\ell} \in \mathbb{R}^{d_e \times d_e}$ is the projection matrix (sometimes called output matrix) of layer $\ell$. In our implementation, we allow this layer to have trainable bias terms too.

Finally, having computed, layer by layer, the hidden outputs $\boldsymbol{X}_{\ell,t}$ for $\ell \in [L]$, we apply an affine classifier (with softmax) to obtain the output of the model

$$\boldsymbol{f}(\boldsymbol{x}) = \mathcal{S}(\widetilde{\boldsymbol{X}}_{L,t} \boldsymbol{W}_{class}^T + \boldsymbol{b}_{class}) \tag{8}$$

This output indicates the probability vector of the next word.

In this paper, we set the dimension of the hidden embeddings $d_e = 768$.

### B.2.2 TRAINING DETAILS

In all of our experiments, we set the learning rate to $5 \times 10^{-5}$, and weight decay to $10^{-4}$. We use a batch size of 512, and train the model for 60k iterations. We use the AdamW optimizer in PyTorch, with 5k iterations of linear warmup, followed by cosine annealing to a learning rate of 0. Each model is trained on a single V100 GPU; the full set of models take around 2 - 3 days to finish training.

### B.3 HIGH-LEVEL REASONING STRATEGY OF THE 3-LAYER TRANSFORMER

We complement the text description of the reasoning strategy of the 3-layer transformer in the main text with Figure 8 below. It not only presents the main strategy of the model, but also summarizes the core evidence for specific parts of the strategy.

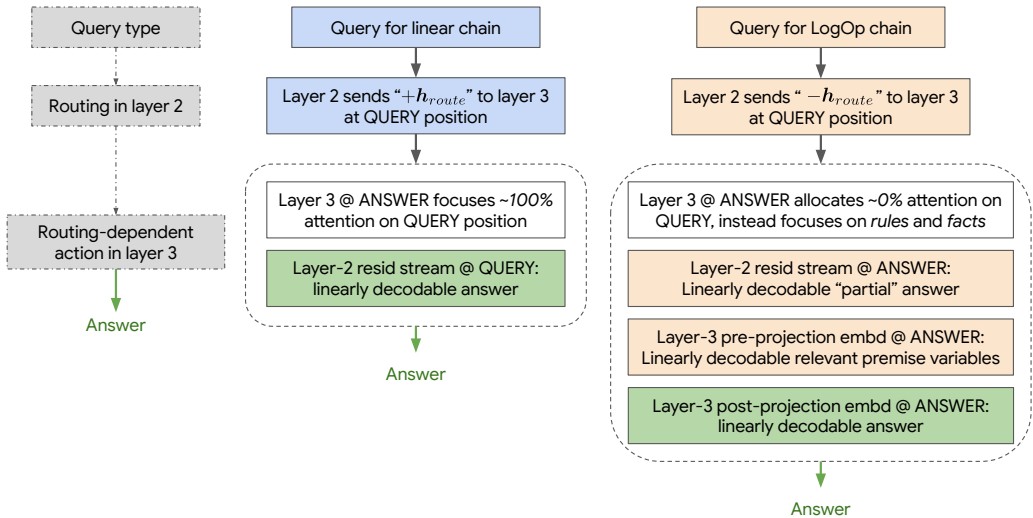

Figure 8: High-level overview of how the 3-layer transformer solves the logic problem. As shown in the grey blocks on the left, the model performs "routing" in layer 2 by sending a routing signal $\boldsymbol{h}_{route}$ to layer 3 (with its "sign" dependent on the query type), then the layer-3 attention block acts according to the "sign" of the routing signal sent to it. The middle (right) chain shows the strategy when the problem queries for linear (LogOp) chain.

### B.4 ANSWER FOR THE LOGOP CHAIN

*Evidence 3a: Distinct behaviors of affine predictors at different layers*. We train two affine classifiers at two positions inside the model (each with 10k samples): $\boldsymbol{W}_{resid,\ell=2}$ at layer-2 residual stream, and $\boldsymbol{W}_{attn,\ell=3}$ at layer-3 attention-block output, both at the position of ANSWER, with the target being the correct first token. In training, if there are two correct answers possible (e.g. OR gate, starting nodes are both TRUE or both FALSE), we randomly choose one as the target; in testing,

we deem the top-1 prediction "correct" if it coincides with one of the answers. We observe the following predictor behavior on the test samples:

1. $\boldsymbol{W}_{attn,\ell=3}$ predicts the correct answer 100% of the time.
2. $\boldsymbol{W}_{resid,\ell=2}$ always predicts one of the variables assigned FALSE (in the fact section) if LogOp is the AND gate, and predicts one assigned TRUE if LogOp is the OR gate.

*Evidence 3b: linearly decodable LogOp information from first two layers.* We train an affine classifier at the layer-2 residual stream to predict the LogOp of the problem instance, over 5k samples (and tested on another 5k samples). The classifier achieves greater than 98% accuracy. We note that training this classifier at the layer-1 residual stream also yields above 95% accuracy.

*Evidence 3c: identification of LogOp-chain starting nodes at layer 3.* Attention heads (3,1) and (3,3), when concatenated, produce embeddings which we can linearly decode the two starting nodes of the LogOp chain with test accuracy greater than 98%. We also find that they focus their attention in the rule section of the context (as shown in Figure 9). Due to causal attention, this means that they determine the two starting nodes from the LogOp-relevant rules. *Remark.* The above pieces of observations suggest the "partial information→refinement" process.[8] To further validate that the embedding from the first two layers are indeed causally linked to the correct answer at the third layer, we perform an activation patching experiment.

*Evidence 3d: linear non-decodability of linear chain's answer.* To provide further contrasting evidence for the linear decodability of the LopOp chain's answer, we experimentally show that it is not possible to linearly decode the answer of the *linear* chain in the model. Due to the causal nature of the reasoning problem (it is only possible to know the answer at or after the QUERY token position), and the causal nature of the decoder-only transformer, we train a set of linear classifiers on all token positions at or after the QUERY token and up to the ANSWER token, and on all layers of the residual stream of the transformer. We follow the same procedure as in Evidence 3c, except in this set of experiments, for contrasting evidence, QUERY is for the LopOp chain, while the classifier is trained to predict the answer of the Linear chain. The maximum test accuracy of the linear classifiers across all aforementioned token positions and layer indices is only 32.7%. Therefore, the answer of the Linear chain is not linearly encoded in the model when QUERY is for the LopOp chain.

*Evidence 3e: layer-2 residual stream at ANSWER is important to correct prediction.* We verify that layer-3 attention does rely on information in the layer-2 residual stream (at the ANSWER position):

- Construct two sets of samples $\mathcal{D}_1$ and $\mathcal{D}_2$, each of size 10k: for every sample $\boldsymbol{X}_{1,n} \in \mathcal{D}_1$ and $\boldsymbol{X}_{2,n} \in \mathcal{D}_2$, the context of the two samples are exactly the same, except the LogOp is flipped, i.e. if $\boldsymbol{X}_{1,n}$ has disjunction, then $\boldsymbol{X}_{2,n}$ has the conjunction operator. If layer 3 of the model has *no* reliance on the $\text{Resid}_{\ell=2}$ (layer-2 residual stream) for LogOp information at the ANSWER position, then when we run the model on any $\boldsymbol{X}_{2,n}$, patching $\text{Resid}_{\ell=2}(\boldsymbol{X}_{n,2})$ with $\text{Resid}_{\ell=2}(\boldsymbol{X}_{n,1})$ at ANSWER should *not* cause significant change to the model's accuracy of prediction. However, we observe the contrary: the accuracy of prediction degrades from 100% to 70.87%, with standard deviation 3.91% (repeated over 3 sets of experiments).

*Observation: LogOp-relevant reasoning at the third layer.* We show that the output from attention heads (3,1) and (3,3) (before the output/projection matrix of the layer-3 attention block), namely $\mathcal{A}_{3,1}(\boldsymbol{X}_2)$ and $\mathcal{A}_{3,3}(\boldsymbol{X}_2)$, when concatenated, contain linearly decodable information about the two starting nodes of the LogOp chain. We frame this as a multi-label classification problem, detailed as follows:

1. We generate 5k training samples and 5k test samples, each of whose QUERY is for the LogOp chain. For every sample, we record the *target* as a 80-dimension vector, with every entry set to 0 except for the two indices corresponding to the two proposition variables which are the starting nodes of the LogOp chain.

2. Instead of placing softmax on the final classifier of the transformer, we use the Sigmoid function. Moreover, instead of the Cross-Entropy loss, we use the Binary Cross-Entropy loss (namely the

---

[8]In fact, the observations suggest that layer 3 performs a certain "matching" operation. Take the OR gate as an example. Knowing which of the three starting nodes (for LogOp and linear chain) are TRUE, and which two nodes are the starting nodes for the LogOp chain are sufficient to determine the first token! This exact algorithm, however, is not fully validated by our evidence; we leave this as part of our future work.

```
torch.nn.functional.binary_cross_entropy_with_logits
```
in PyTorch, which directly includes the Sigmoid for numerical stability).

3. We train an affine classifier, with its input being the concatenated $\text{Concat}[\mathcal{A}_{3,1}(\boldsymbol{X}_2), \mathcal{A}_{3,3}(\boldsymbol{X}_2)]$ (a 512-dimensional vector) on every training sample, and with the targets and training loss defined above. We use a constant learning rate of $0.5 \times 10^{-3}$, and weight decay of $10^{-2}$. The optimizer is AdamW in PyTorch.

4. We assign a "correct" evaluation of the model on a test sample only if it correctly outputs the two target proposition variable as the top-2 entries in its logits. We observe that the classifier achieves greater than 98% once it converges.

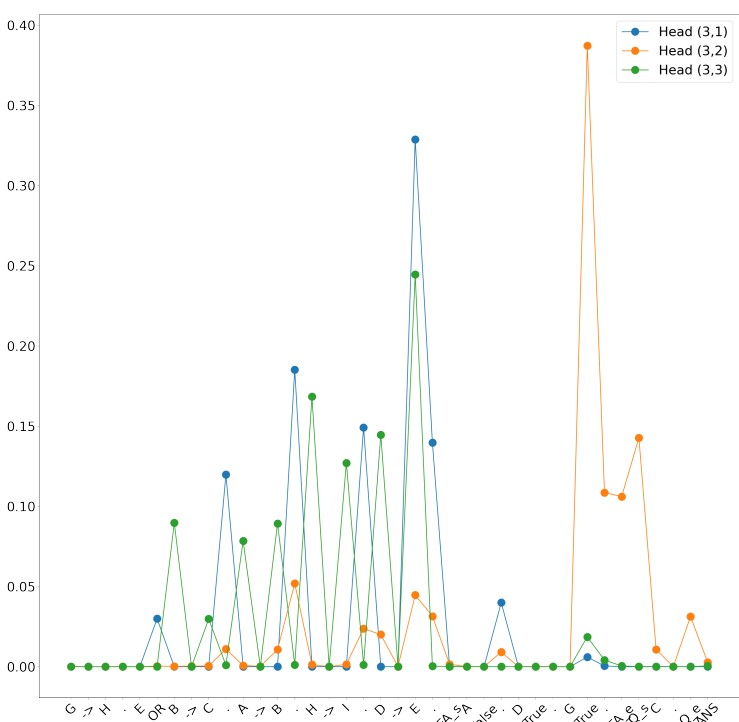

Figure 9: Attention statistics, averaged over 500 samples, all of which query for the LogOp chain. The x-axis is simply an example prompt that helps illustrate where the attention is really placed at. Observe that only attention head (3,2) pays significant attention to the fact section. The other two heads focus on the rule section. Note that none of them concentrate attention on the QUERY token. Reminder: due the the design of the problem, the rule, fact and query sections all have consistent length for every sample!

## B.5 EXTRA REMARKS

**Observation 3 supplement: linearly-decodable linear-chain answer at layer 2**. We simply frame the learning problem as a linear classification problem. The input vector of the classifier is the same as the input to the layer-3 self-attention block, equivalently the layer-2 residual-stream embedding. The output space is the set of proposition variables (80-dimensional vector). We train the classifier on 5k training samples (all whose QUERY is for the linear chain) using the AdamW optimizer, with learning rate set to $5 \times 10^{-3}$ and weight decay of $10^{-2}$. We verify that the trained classifier obtains an accuracy greater than 97% on an independently sampled test set of size 5k (all whose QUERY is for the linear chain too).

**Remarks on truth value determination**. Evidence suggests that determining the truth value of the simple propositional logic problem is easy for the model, as the truth value of the final answer is linearly decodable from layer-2 residual stream (with 100% test accuracy, trained on 10k samples) when we give the model the context+chain of thought right before the final truth value token. This

is expected, as the main challenge of this logic problem is not about determining the query's truth value, but about the model spelling out the minimal proof with careful planning. When abundant CoT tokens are available, it is natural that the model knows the answer even in its second layer.

## C  THE REASONING CIRCUIT IN MISTRAL-7B: EXPERIMENTAL DETAILS

### C.1  PROBLEM FORMAT

We present six examples of the propositional-logic problem in context to the Mistral-7B model, and ask for its answer to the seventh problem. An example problem is presented below.

```
Rules: Z or F implies B. D implies C.
Facts: D is true. Z is true. F is false.
Question: state the truth value of C.
Answer: D is true. D implies C; C is true.
Rules: U implies Y. G or I implies Q.
Facts: I is true. U is true. G is false.
Question: state the truth value of Y.
Answer: U is true. U implies Y; Y is true.
Rules: G or Z implies E. U implies K.
Facts: U is true. G is true. Z is false.
Question: state the truth value of E.
Answer: G is true. G or Z implies E; E is true.
Rules: G implies U. Y or A implies V.
Facts: Y is true. G is true. A is false.
Question: state the truth value of V.
Answer: Y is true. Y or A implies V; V is true.
Rules: U implies W. H or B implies L.
Facts: B is false. U is true. H is true.
Question: state the truth value of W.
Answer: U is true. U implies W; W is true.
Rules: F or A implies Y. E implies I.
Facts: A is false. F is true. E is false.
Question: state the truth value of Y.
Answer: F is true. F or A implies Y; Y is true.
Rules: B or F implies D. S implies T.
Facts: S is true. F is true. B is false.
Question: state the truth value of T.
Answer:
```

*Remark.* To ensure fairness to the LLM, we balance the number of in-context examples which queries the OR chain and the linear chain: each has 3 in-context examples. The order in which the in-context examples are presented (i.e. the order in which the examples with OR or linear-chain answer) is random. Please note that, in the six in-context examples, we do allow the truth value assignment for the premise variable of the linear chain to be FALSE when this chain is not being queried, however, the actual question (the seventh example which the model needs to answer) always sets the truth value assignment of the linear chain to be TRUE, so the model cannot take a shortcut and bypass the "QUERY→Relevant Rule" portion of the reasoning path.

Additionally, when reporting the accuracy of the model being above 70% in the main text, we are querying the model for the LogOp and linear chain with 50% probability respectively. More precisely, we test the model on 400 samples, and we find that the model has 96% accuracy when QUERY is for the linear chain, and 70% accuracy when QUERY is for the OR chain (so they average above 70% accuracy).

### C.2  CAUSAL MEDIATION ANALYSIS: FURTHER EXPLANATIONS

This subsection complements the causal mediation analysis methodologies we presented in §4.2 in the main text. In particular, we aim to visualize how the interventions are done in the circuit discovery and verification processes, by using a 2-layer 2-head transformer as an example for simplicity.

Figure 10 illustrates the activation patching procedure of circuit discovery. Recall that we are studying how a component inside the transformer causally influences the output of the model. In the specific example, we show how we would examine the causal influence of attention head (0,2)'s activations on the correct inference of the model.

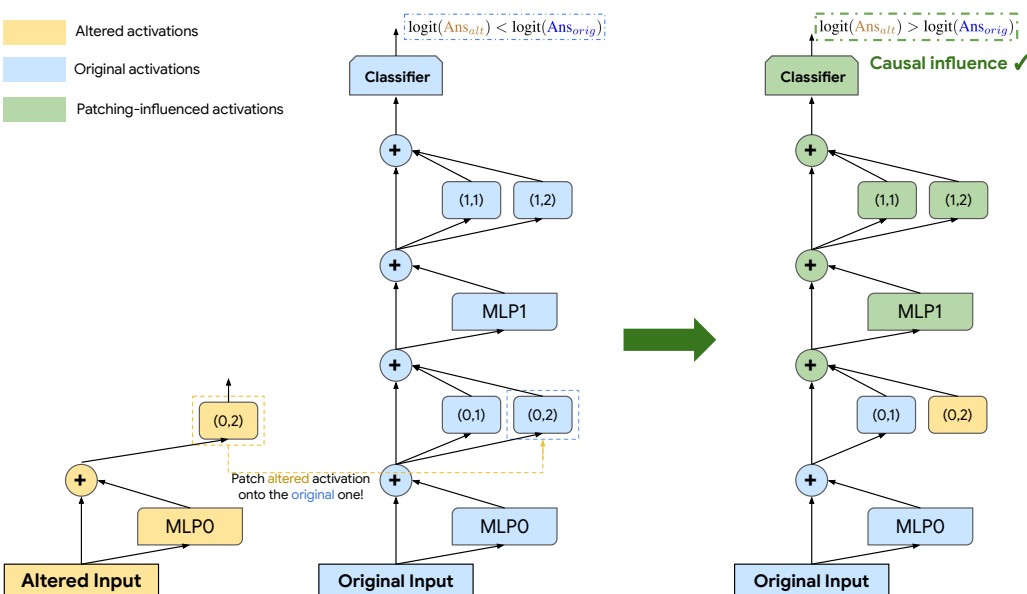

Figure 10: Illustration of how activation patching is performed in the *circuit-discovery* process (necessity-based patching). We use a 2-layer 2-head transformer as a simplified example here, and use $(\ell, h)$ to denote an attention head.

In this specific illustrated example, we are studying the causal influence of attention head (0,2)'s activation on the correct inference of the model. After caching the altered activations of (0,2) (shown on the left), we run the model on the original prompt and cache the activations (shown in the middle), then replace the original activation of head (0,2) by its altered activations, and let the rest of the layers be computed normally (shown on the right) — they now operate out of distribution, and are colored in green.

In this specific example, the intervened run outputs logits which reflect "belief altering": that is, the probability for the answer token of the original prompt now is lower than the answer token for the altered prompt. This indicates that head (0,2) has causal influence on the corrent inference of the model.

Circuit verification, on the other hand, goes through a somewhat more complex process of interventions, as illustrated in Figure 11. Recall our main procedure (discussed in the main text).

1. We run the LLM on the *original* prompts, and cache the activations of the attention heads.

2. Now, we run the LLM on the corresponding *altered* prompts, however, we *freeze* all the attention heads' activations inside the model to their activations on the *original* prompts, *except* for those in the circuit $\mathcal{C}$ which we wish to verify (i.e. only the attention heads in $\mathcal{C}$ are allowed to run normally). We record the (circuit-intervened) altered logit differences on the altered prompts.

3. We average the circuit-intervened altered logit differences across the samples, namely $\frac{1}{N} \sum_{n=1}^{N} \Delta_{orig \to alt}^{\mathcal{C}}$[9], and check whether they approach the "maximal" altered logit difference, namely $\Delta_{alt}$[10].

---

[9]This specific term reflects, on average, how much the model favors outputting the answer token for the altered prompts over the original prompts *after* the circuit interventions.

[10]Recall that this term is obtained by running the LLM on the altered prompts without any modification to its internal activations at all. This specific term reflects, on average, how much the (un-intervened) model favors outputting the answer tokens for the altered prompts over those of the original prompts, when it is run on the altered prompts.

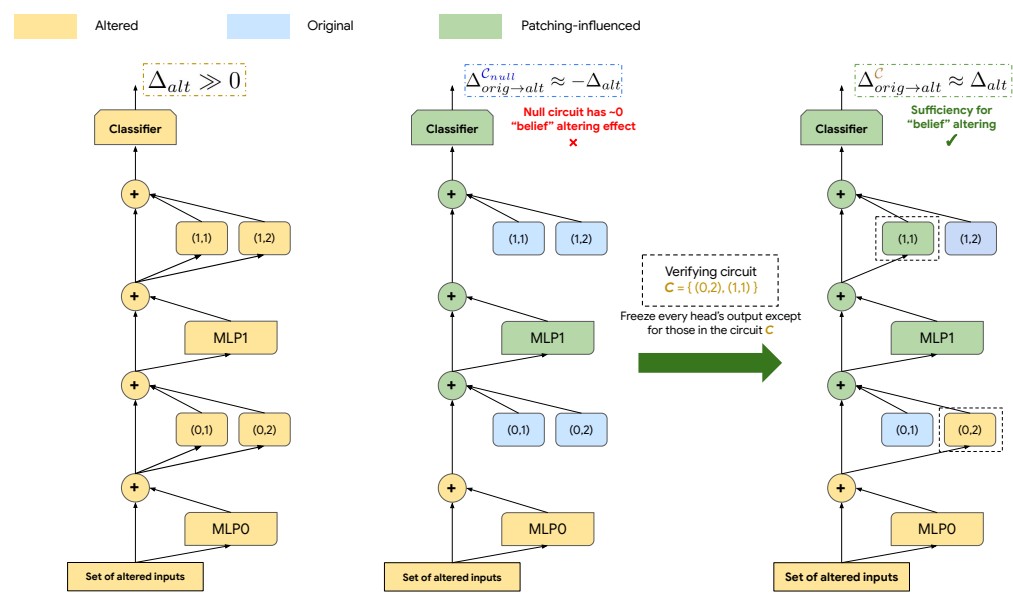

Figure 11: Illustration of how activation patching is performed in the *circuit-verification* process (sufficiency-based patching). We use a 2-layer 2-head transformer as a simplified example here. We use $(\ell, h)$ to denote an attention head.

In this specific illustrated example, we are verifying whether the circuit $\mathcal{C} = \{(0, 2), (1, 1)\}$ consisting of the two attention heads is *sufficient* for altering the "belief" of the model.

We first obtain $\Delta_{alt}$, the average altered logit difference, by running the model on the altered prompts without interventions. We also run the model on the original prompts and cache the attention heads' activations (these "original" activations are colored in blue in the figure).

The naive "baseline" for sufficiency verification is the null circuit $\mathcal{C}_{null} = \emptyset$ (shown in the middle): we freeze all the attention heads to their original activations when running the model on the altered prompts. This null circuit, as shown in this example, barely alters the model's "belief" from the original, as $\Delta_{orig \to alt}^{\mathcal{C}_{null}} \approx -\Delta_{alt}$, i.e. the model still strongly favors outputting the answer tokens for the original prompts over those of the altered prompts on average.

In contrast, if we unfreeze the attention heads in the circuit $\mathcal{C}$ when running the model (shown on the right), we observe that the model's circuit-intervened logit difference approaches the "maximal" altered logit difference $\Delta_{alt}$. This indicates that the attention heads in $\mathcal{C}$ are sufficient for correctly manipulating the information flow (and processing the information) for reaching the right answer.

*Remark.* As the reader can observe, we do *not* freeze the MLPs in our intervention experiments. We note that the MLPs do not move information between the residual streams at different token positions, as they only perform processing of whatever information present at the residual stream. Therefore, similar to Wang et al. (2023), we consider the MLPs as part of the "direct" path between two attention heads, and allow information to flow freely through them, instead of freezing them and disrupting the information flow between attention heads.

## C.3    FINER DETAILS OF QUERY-BASED ACTIVATION PATCHING

In this subsection, we present and visualize the attention heads with the highest average intervenes logit differences, along with their standard deviations (error bars).

### C.3.1    QUERY-BASED ACTIVATION PATCHING EXPERIMENTS: METRICS

We rely on a calibrated version of the logit-difference metric often adopted in the literature for the QUERY-based activation patching experiments (aimed at keeping the score's magnitude between 0

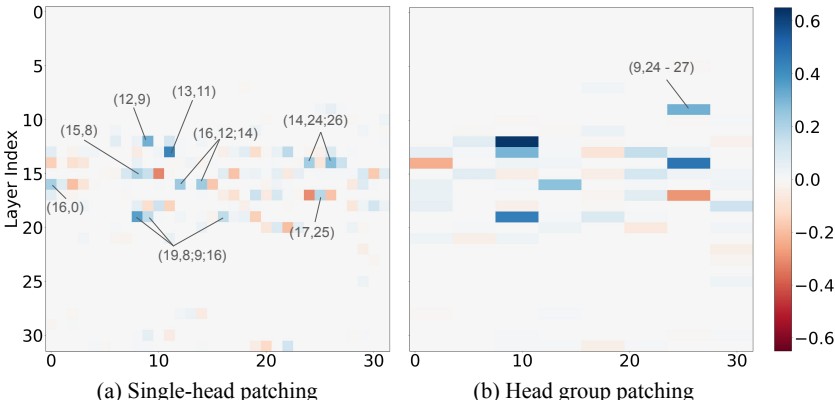

(a) Single-head patching        (b) Head group patching

Figure 12: Attention head patching, highlighting the ones with the highest intervened logit difference; $x$-axis is the head index. (a) shows single-head patching results (same as the one shown in the main text, repeated here for the reader's convenience), and (b) shows a coarser-grained head patching in groups. In (b), we only highlight the head groups that are not captured well by (a).

and 1). In particular, we compute the following metric for head $(\ell, h)$ at token position $t$:

$$\frac{\frac{1}{N} \sum_{n \in [N]} \Delta_{orig \to alt, n; (\ell, h, t)} - \Delta_{orig}^{\dagger}}{\Delta_{alt} - \Delta_{orig}^{\dagger}}. \tag{9}$$

where $\Delta_{orig}^{\dagger} = \frac{1}{N} \sum_{n \in [N]} \text{logit}(\boldsymbol{X}_{orig,n})[y_{alt,n}] - \text{logit}(\boldsymbol{X}_{orig,n})[y_{orig,n}]$, and $\Delta_{alt} = \frac{1}{N} \sum_{n \in [N]} \text{logit}(\boldsymbol{X}_{alt,n})[y_{alt,n}] - \text{logit}(\boldsymbol{X}_{alt,n})[y_{orig,n}]$. The closer to 1 this score is, the stronger the model's "belief" is altered; the closer to 0 it is, the closer the model's "belief" is to the original unaltered one.

Each of our experiments are done on 60 samples unless otherwise specified — we repeat some experiments (especially the attention-head patching experiments) to ensure statistical significance when necessary.

### C.3.2 ATTENTION HEAD GROUP PATCHING

We note that *Grouped-Query Attention* used by Mistral-7B adds subtlety to the analysis of which attention heads have strong causal influence on the LLM's correct output. (In Mistral-7B-v0.1, each attention layer has 8 key and value activations, and 32 query activations. Therefore, heads $(\ell, h \times 4)$ to $(\ell, h \times 4 + 3)$ share the same key and value activation.) Patching a single head might not yield a high logit difference, since other heads in the same group (which possibly perform a similar function) could overwhelm the patched head and maintain the model's previous "belief". Therefore, we also run a *coarser-grained* experiment which simultaneously patches the attention heads sharing the same key and value activations, shown in Figure 12(b). This experiment reveals that heads belonging to the group (9, 24 - 27) also have high intervened logit difference. Combining with the observation that (9,25;26) have somewhat positive scores in the single-head patching experiments, and by examining these two head's attention patterns (which shall be discussed in detail in the immediate next subsection), we determine that they also should be included in the circuit.

### C.3.3 ATTENTION PATTERNS OF QUERY-SENSITIVE ATTENTION HEADS

In this subsection, we provide finer details on the attention patterns of the attention heads we discovered in Section 4.3.1. Note that the attention weights percentage we present in this section are calculated by dividing the observed attention weight at a token position by the total amount of attention the head places in the relevant context, i.e. the portion of the prompt which excludes the 6 in-context examples.

**Queried-rule locating heads**. Figure 13 presents the average attention weight the queried-rule locating heads place on the "conclusion" variable and the period "." immediately after the queried

rule at the QUERY token position (i.e. the query activation of the heads come from the residual stream at the QUERY token position) — (12,9) is an exception to this recording method, where we only record its weight on the conclusion variables alone, and already observe very high weight on average. The heads (12,9), (14,24), (14,26), (9,25), (9,26) indeed place the majority of their attention on the correct position *consistently* across the test samples. The reason for counting the period after the correct conclusion variable as "correctly" locating the rule is that, it is known that LLMs tend to use certain "register tokens" to record information in the preceding sentence.

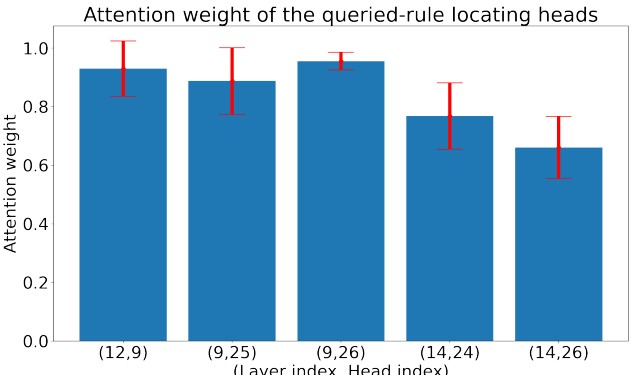

Figure 13: Average attention weights of the queried-rule locating heads, along with the standard deviations. The weights are calculated by dividing the actual attention weight placed on the correct "conclusion" variable of the rule and the period "." immediately after, by the total amount of attention placed in the relevant context (i.e. the prompt excluding the 6 in-context examples). *Head (12,9) is an exception: we only record its attention right on the conclusion variable, and still observe $93.0 \pm 9.4\%$ "correctly placed" attention on average.*

We can observe that head (12,9) has the "cleanest" attention pattern out of the ones identified, placing on average $93.0 \pm 9.4\%$ of it attention on the correct conclusion variable alone. The more diluted attention patterns of the other heads likely contribute to their weaker intervened logit difference score shown in §4.3.1 in the main text.

**Queried-rule mover heads**. Figure 14 shows the attention weight of the queried-rule mover heads. While they do not place close to 100% attention on the QUERY location consistently (when the query activation comes from the residual stream from token ":", right before the first answer token), the top-1 attention weight consistently falls on the QUERY position, and the second largest attention weight is much smaller. In particular, head (13,11) places $54.2 \pm 12.5\%$ attention on the QUERY position on average, while the second largest attention weight in the relevant context is $5.2 \pm 1.1\%$ on average (around 10 times smaller; *this ratio is computed per sample and then averaged*).

**Extra note about head (16,0)**: it does *not* primarily act like a "mover" head, as its attention statistics suggest that it processes an almost even *mixture* of information from the QUERY position and the ":" position. Therefore, while we present its statistics along with the other queried-rule mover heads here since it does allocate significant attention weight on the QUERY position on average, we do not list it as such in the circuit diagram of Figure 3. **Furthermore, we do not include it as part of the circuit $\mathcal{C}$ in our circuit verification experiments**.

**Fact processing heads**. Figure 15 below shows the attention weights of the fact processing heads; the attention patterns are obtained at the ":" position, right before the first answer token, and we sum the attention weights in the Fact section (starting at the first fact assignment, ending on the last "." in this section of the prompt). It is clear that they place significant attention on the Fact section of the relevant context. Additionally, across most samples, we find that these heads exhibit the tendency to assign lower amount of attention on the facts with FALSE value assignments across most samples, and on a nontrivial portion of the samples, they tend to place greater attention weight on the correct fact (this second ability is not consistent across all samples, however). Therefore, they do appear to perform some level of "processing" of the facts, instead of purely "moving" the facts to the ":" position.

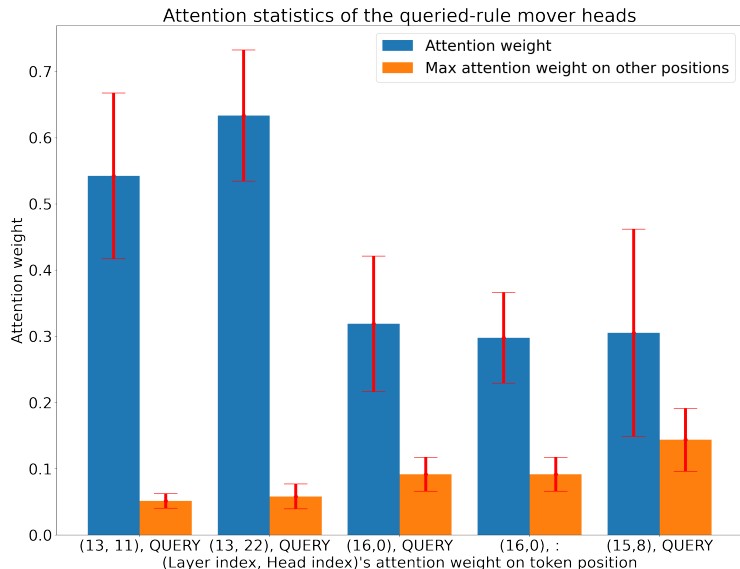

Figure 14: Average attention weights of the queried-rule mover heads, along with the standard deviations. The raw attention patterns are obtained at token position ":" (i.e. the query activation comes from the residual stream at the ":" position), right before the first answer token, and the exact attention weight (indicated by the blue bars) is taken at the QUERY position; for head (16,0), we also obtain its attention weight at the ":" position, as we found that it also allocates a large amount of attention weight to this position in addition to the QUERY position. Note: for (15,8), we found that it only acts as a "mover" head when the linear chain is being queried, so we are only reporting its attention weight statistics in this specific scenario; the other heads do not exhibit this interesting behavior, so we report those heads' statistics in all query scenarios.

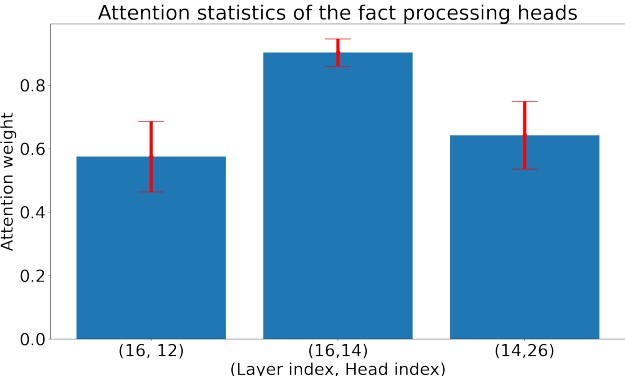

Figure 15: Average attention weights of the fact processing heads computed at the ":" token position (last position before the answer), along with the standard deviations. The weights are calculated by dividing the actual attention weight placed in the Fact section by the total amount of attention placed in the relevant context (i.e. the part of the prompt excluding the 6 in-context examples).

**Decision heads**. Figure 16 shows the attention weights of the decision heads *on samples where the model outputs the correct answer (therefore, about 70% of the samples)*. The attention patterns are obtained at the ":" position. We count the following token positions as the "correct" positions:

- In the Rules section, we count the correct answer token and the token immediately following it as correct.

- In the Facts section, we count the sentence of truth value assignment of the correct answer variable as correct (for example, "A is true.").

- Note: the only exception is head (19,8), where we only find its attention on exactly the correct tokens (not counting any other tokens in the context); we can observe that it still has the cleanest attention pattern for identifying the correct answer token.

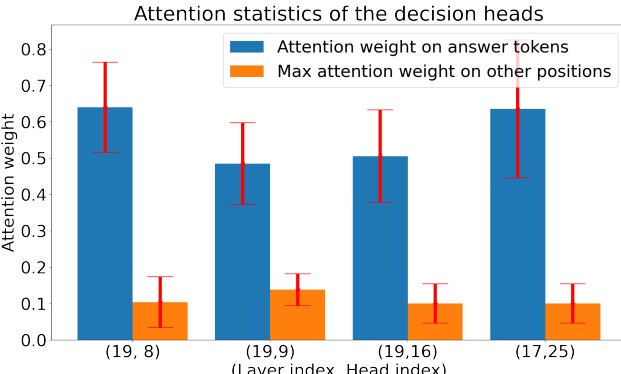

Figure 16: Average attention weights of the decision heads, along with the standard deviations. The weights are calculated by dividing the actual attention weight placed on the correct answer tokens by the total attention the model places in the relevant context.

An interesting side note worth pointing out is that, (17,25) tends to only concentrate its attention in the facts section, similar to the fact-processing heads. The reason which we do not classify it as a fact-processing head and instead as a decision head is that, in addition to finding that their attention patterns tend to concentrate on the correct fact, evidence presented in §C.6 below suggest that they are not directly responsible for locating and moving the facts information to the ":" position, while the heads (16,12;14) exhibit such tendency strongly.

### C.4 SUFFICIENCY TESTS FOR CIRCUIT VERIFICATION

In §4.3.2, we presented a sufficiency test of the circuit. Here, we elaborate further on the experimental procedures and finer details of the experiment.

The circuit which we perform verification on is the union of the four attention head families, $\mathcal{C} = QRLH \cup QRMH \cup FPH \cup DH$, with

- $QRLH$ = Queried-Rule Locating Heads = $\{(9, 25; 26), (12, 9), (14, 24; 26)\}$ patched at token position QUERY;
- $QRMH$ = Queried-Rule Mover Heads = $\{(13, 11; 22), (15, 8)\}$ patched at the ":" position (the last position of context);
- $FPH$ = Fact-Processing Heads = $\{(16, 12; 14), (14, 26)\}$ patched at the ":" position;
- $DH$ = Decision Heads = $\{(19, 8; 9; 16), (17, 25)\}$ patched at the ":" position.

An exception is that the queried-rule locating head $(14, 24)$ is also patched at the ":" position, as we observed that it tends to concentrate attention at the queried rule at this position: it does not locate the queried rule as consistently as it does at the QUERY position, however. We still chose to patch it at this position as we found that it tends to improve the altered logit difference, indicating that either the model relies on this head to pass certain additional information about the queried rule to the ":" position, or certain later parts of the circuit do rely on this head for queried-rule information. The exact function of this attention head remains part of our future study in the reasoning circuit of Mistral-7B. We likely need to examine this head's role in other reasoning problems to clearly understand what its role is at different token positions, and whether there is deeper meaning behind the fact that, their apparently redundant actions at different token positions all seem to have causal influence on the model's inference.

**Challenges of reasoning circuit sufficiency verifications**. From what we can see, verifying the sufficiency of a *reasoning* circuit is a major open problem. Part of the root of the problem lies in what

exactly counts as a circuit that is truly relevant to *reasoning*: attention heads and MLPs responsible for lower-level processing such as performing change of basis of the token representations, storing information at register tokens (such as the periods "." after sentences), and so on, do not truly belong to a "reasoning" circuit in the narrow definition of the term. In our considerations, a "narrow" definition of a reasoning circuit is one which is *QUERY sensitive* and *has strong causal influence on the correct output of the model on the reasoning problems*. The first condition of QUERY sensitivity is justified by noting that the QUERY lies at the root of the reasoning chain of "QUERY→Relevant Rule(s)→Relevant Fact(s)→Decision". We do not analyze through what circuit/internal processing the "QUERY", "Relevant Rule(s)" and "Relevant Fact(s)" underwent from token level to representation level (notice that the reasoning circuit we identified starts at layer 9: it is entirely possible for the token embeddings of these important items to have undergone significant processing by the attention heads and MLPs in the lower layers). Simply setting the lower layers' embeddings to the zero vector, to their mean activations or some fixed embeddings which erase the instance-dependent information could completely break the circuit.

### C.5 QUERIED-RULE LOCATION INTERVENTIONS: ANALYZING THE QUERIED-RULE LOCATING HEADS

In this experiment, we *only* swap the *location* of the linear rule with the LogOp rule in the *Rule* section of the question, while keeping everything else the same (including all the in-context examples). As an example, we alter "Rules: A or B implies C. D implies E." to "Rules: D implies E. A or B implies C." while keeping everything else the same. The two prompts have the same answer.

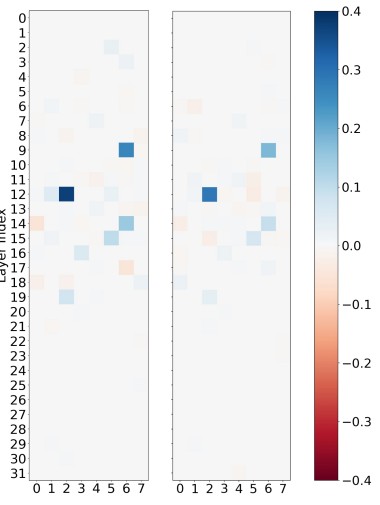

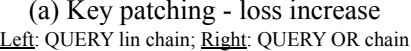

(a) Key patching - loss increase
Left: QUERY lin chain; Right: QUERY OR chain

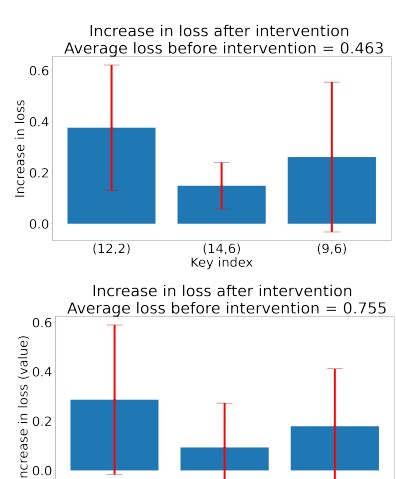

(b) Average increase in loss
Top: QUERY lin chan; Bottom: QUERY OR chain

Figure 17: Key activations patching results. In this experiment, we swap the location of the linear rule and the LogOp rule in the Rule section and keep everything else in the prompt the same; we patch the key activations of the attention heads in the Rule section only. (a) visualizes the average increase in the cross-entropy loss with respect to the true target (the true first token of the answer) for all key indices, and (b) shows the average and standard deviation of the top three key indices with the highest loss increase. Observe that these are the keys for the queried-rule locating heads (12,9), (14,24;26) and (9,25;26) identified in §4.3.1.

If the *queried-rule locating heads* (with heads (12,9), (14,25;26), (9,25;26) being the QUERY-sensitive representatives) indeed perform their functions as we described, then when we run the model on the clean prompts, patching in the *altered key activations* at these heads (within the Rules section) should cause "negative" change to the model's output, since it will cause these heads to mistake the queried-rule location in the altered prompt to be the right one, consequently storing the wrong rule information at the QUERY position. In particular, the model's *cross-entropy loss* with respect to the original target should increase. This is indeed what we observe.

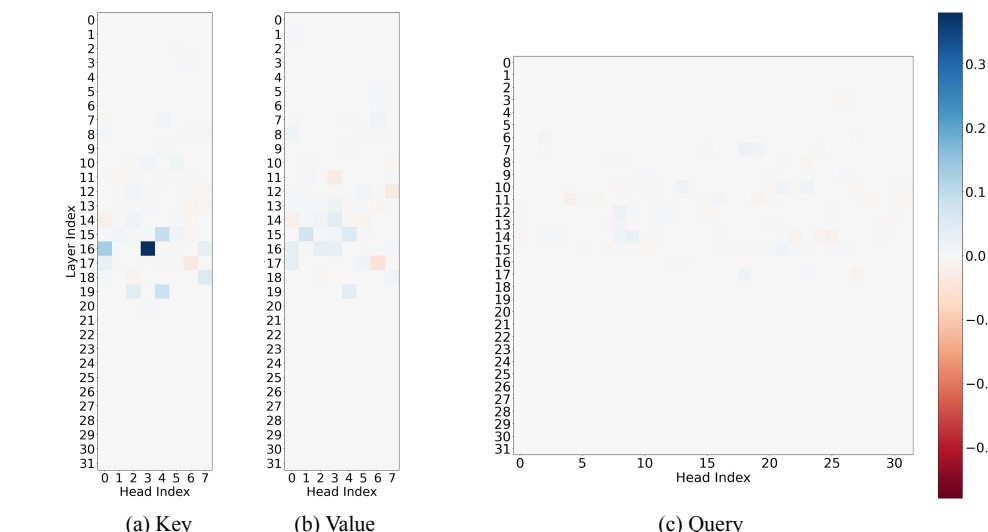

(a) Key      (b) Value      (c) Query

Figure 18: Key, value and query activation patching in the Facts section, with the metric being the calibrated intervened logit difference. The truth value assignments for the OR chain is flipped (while keeping everything else in the prompt the same), and the OR chain is always queried. Observe that only the *key* activations at index (16,3) obtain a high intervened logit difference score of approximately 0.34 (this key index corresponds to the attention heads (16, 12 - 15)). Also observe that the value and query activations in the facts section do not exhibit strong causal influence on the correct inference of the model.

The average increase in cross-entropy loss exhibit a trend which corroborate the hypothesis above, shown in Figure 17. While the average cross-entropy loss on the original samples is 0.463, patching (12,9), (14,24;26) and (9,25;26)'s keys (with corresponding key indices (12,2), (14,6) and (9,6)) in the Rule section. Patching the other *QUERY-sensitive* attention heads' keys in the Rule section, in contrast, show significantly smaller influence on the loss on average, telling us that their responsibilities are much less involved with *directly* finding or locating the queried rule via attention.

Note: this set of experiments was run on 200 samples instead of 60, since we noticed that the standard deviation of some of the attention heads' loss increase is large.

*Remark.* While attention heads with key index (15,5) (i.e. heads (15, 20-23)) did not exhibit nontrivial sensitivity to QUERY-based patching (discussed in Section 4.3.1 in the main text), patching this key activation does result in a nontrivial increase in loss. Examining the attention heads belonging to this group, we find that they indeed also perform the function of locating the queried rule similar to head (12,9). We find them to be less accurate and place less attention on the exact rule being queried on average, however: this weaker "queried-rule locating ability" likely contributed to their low scores in the QUERY-based patching experiments presented in the main text.

### C.6 FACTS INTERVENTIONS: ANALYZING THE FACT-PROCESSING AND DECISION HEADS

In this section, we aim to provide further validating evidence for the fact-processing heads and the decision heads. We experiment with flipping the truth value assignment for the OR chain while keeping everything else the same in the prompt (we always query for the OR chain in this experiment). As an example, we alter "Rules: A or B implies C. D implies E. Facts: **A is true. B is false.** D is true. Query: please state the truth value of C." to "Rules: A or B implies C. D implies E. Facts: **A is false. B is true.** D is true. Query: please state the truth value of C.". In this example, the answer is flipped from A to B. The (calibrated) intervened logit difference is still a good choice in this experiment, therefore we still rely on it to determine the causal influence of attention heads on the model's inference, just like in the QUERY-based patching experiments.

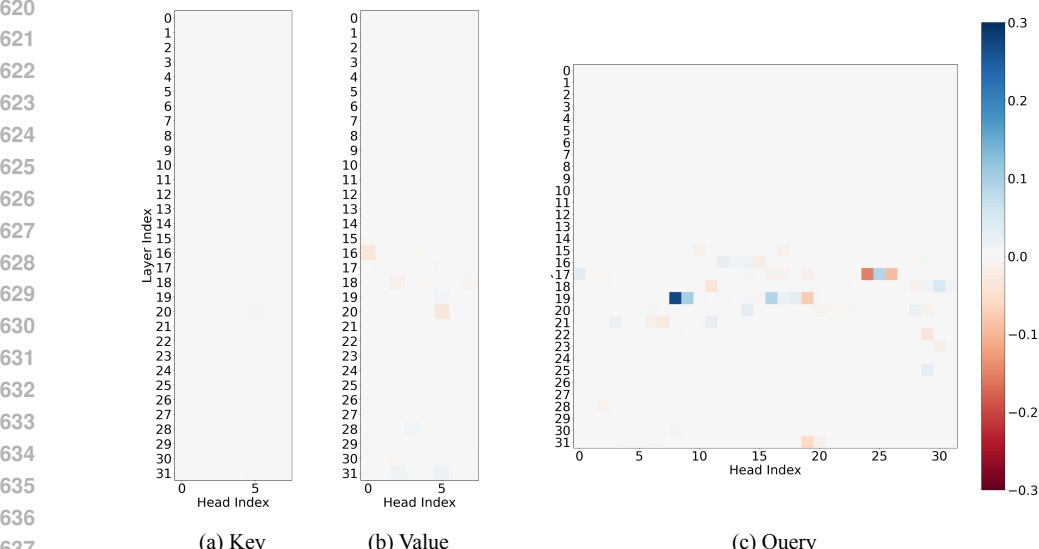

(a) Key          (b) Value          (c) Query

Figure 19: Key, value and query activation patching at the ":" position (last token position in the context, right before the answer token), with the metric being the calibrated intervened logit difference. The truth value assignments for the OR chain is flipped (while keeping everything else in the prompt the same), and the OR chain is always queried. Observe that only the *query* activations at index (19,8) obtain a high intervened logit difference score of approximately 0.28; the other decision heads (19,9;16) and (17,25) also obtain nontrivial scores when their queries are patched. Also observe that the key and value activations at the ":" position do not exhibit strong causal influence on the correct inference of the model when we only flip the truth value assignments for the OR chain.

If the *fact-processing heads* (with (16,12;14) being the QUERY-sensitive representatives) indeed perform their function as described (moving and performing some preliminary processing of the facts as described before), then patching the *altered key activations in the Facts section* of the problem's context would cause these attention heads to obtain a nontrivial intervened logit difference, i.e. it would help in bending the model's "belief" in what the facts are (especially the TRUE assignments in the facts section), thus pushing the model to flip its first answer token. This is indeed what we observe. In Figure 18, we see that only the key activations with index (16,3) (corresponding to heads (16, 12 - 15)) obtain a much higher score than every other key index, yielding evidence that only the heads with key index (16,3) rely on the facts (especially the truth value assignments) for answer. Moreover, notice that patching the *key* activations of the *decision heads* does not yield a high logit difference on average, telling us that the decision heads do *not directly* rely on the *truth value assignment* of the variables for inference (we wish to emphasize again that, the *positions* of the variables in the Facts section are not altered, only the truth value assignments for the two variables of the OR chain are flipped).

Finally, for additional insights on the decision heads (19,8;9;16) and (17,25), we find that by patching the query activations of these decision heads at the ":" position yields nontrivial intervened logit difference, as shown in Figure 19(c) ((19,8) has an especially high score of about 0.27). In other words, the query activation at the ":" position (which should contain information for flipping the answer from one variable of the OR chain to the other, as gathered by the fact-processing heads) being fed into the decision heads indeed have causal influence on their "decision" making. Moreover, patching the value activation of these heads at ":" does not yield nontrivial logit difference, further suggesting that it is their attention patterns (dictated by the query information fed into these heads) which influence the model's output logits.

### C.7 EARLY EVIDENCE OF A SIMILAR REASONING CIRCUIT IN GEMMA-2-9B

In this section, we present a preliminary analysis of the reasoning circuit of Gemma-2-9B in solving the same reasoning problem which Mistral-7B was examined on from before. *We find that the discovered attention heads' attention patterns inside Gemma-2-9B bear surprising resemblance to Mistral-7B's*: according to their highly specialized attention patterns, they can also be categorized into the four families of attention heads which Mistral-7B employs to solve the problem, namely the queried-rule locating heads, queried-rule mover heads, fact-processing heads, and decision heads. While it is too early to draw precise conclusions on how similar the two circuits in the two LLMs truly are, the preliminary evidence suggests that the reasoning circuit we found in this work potentially has some degree of universality.

Additionally, we caution the reader that the experimental study in this subsection is less exhaustive in nature compared to our study of Mistral-7B, due to limitations in our computation budget.

#### C.7.1 QUERY-BASED ATTENTION HEAD ACTIVATION PATCHING

We perform activation patching of the attention head output of Gemma-2-9B, by flipping the QUERY in the prompt pairs. This is the same procedure we used to discover the attention head circuit for Mistral-7B as discussed in §4.2 and C.2. We highlight the attention heads with the strongest causal influence on the model's (correct) inference in Figure 20.

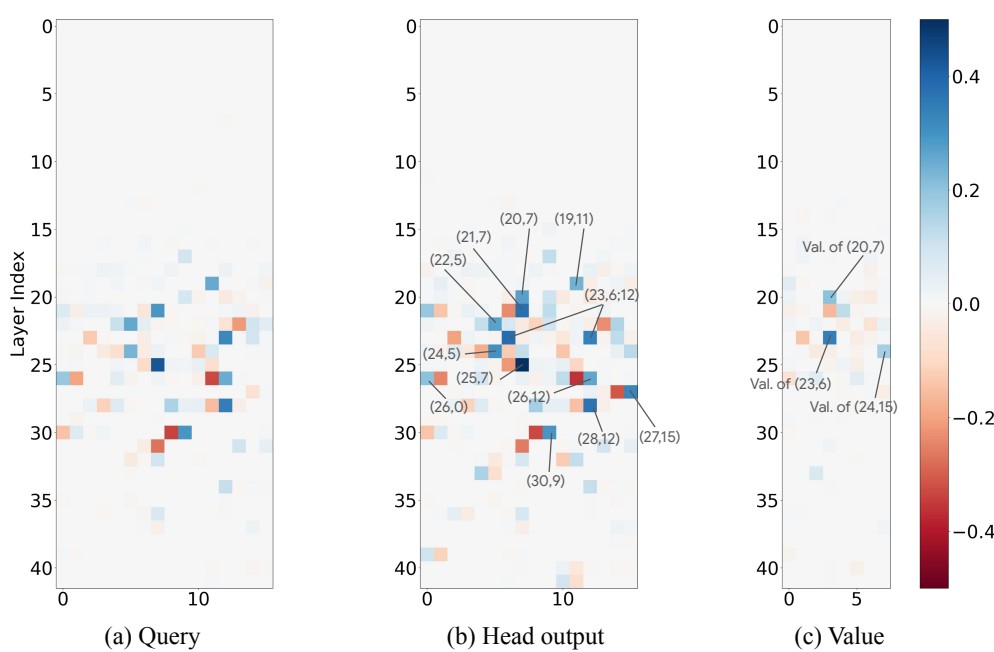

Figure 20: QUERY-based activation patching results of Gemma-2-9B, with sub-component patching on the query and value activations. We highlight the attention heads with the highest calibrated intervened logit difference.

#### C.7.2 ATTENTION PATTERNS OF QUERY-SENSITIVE ATTENTION HEADS IN GEMMA-2-9B

**Queried-rule locating heads**. The queried rule locating heads inside Gemma-2-9B, namely $\{(19, 11), (21, 7), (22, 5), (23, 12)\}$, are very similar in their attention patterns to those in Mistral-7B. At the QUERY position, their attention concentrates on the conclusion token of the queried rule, and the "." which follows. Interestingly, heads (21,7), (22,5) and (23,12) also tend to place some attention on the "implies" token of the queried rule. Another intriguing difference they exhibit is redundant behavior: these attention heads are often observed to have almost exactly the same attention pattern at the "." and "Answer" token positions following the QUERY token. We visualize their attention statistics at the QUERY position in Figure 21.

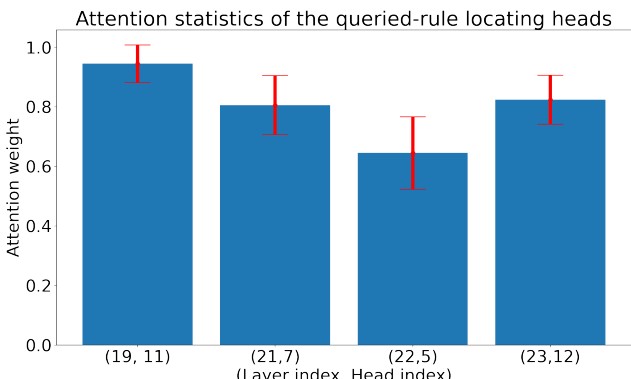

Figure 21: Average attention weights of the queried-rule locating heads in Gemma-2-9B, along with the standard deviations. The attention pattern is obtained at the QUERY position (i.e. query activation of the attention head is from the residual stream at the QUERY token position). We record the attention weight on the queried rule.

**Queried-rule mover heads.** When the query activations of the queried-rule mover heads $\{(20, 7), (23, 6), (24, 15)\}$ come from the ":" residual stream, they have fixed attention patterns which focus a large portion of their attention weights on the QUERY token and two token positions following it, namely the "." and "Answer" token. Their attention weights are slightly more diffuse compared to their counterparts in Mistral-7B, likely due to the queried-rule locating heads performing similar functions at the "." and "Answer" positions. Furthermore, as shown in Figure 20(c), we note that these attention heads are the only ones where patching their value activations results in a large intervened logit difference, further suggesting their role in performing a fixed "moving" action. We record their attention weights in Figure 22.

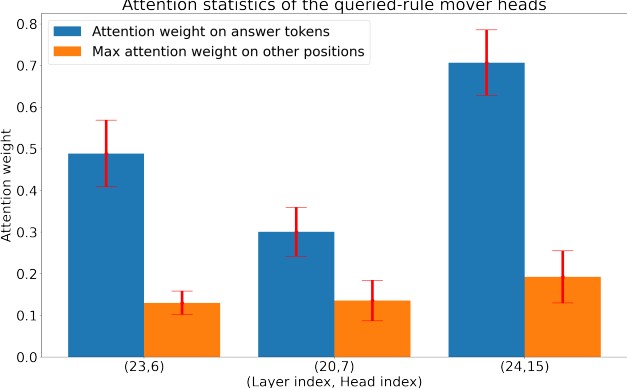

Figure 22: Average attention weights of the queried-rule mover heads in Gemma-2-9B, along with the standard deviations. The attention pattern is obtained at the ":" position, and we sum the attention weights at the QUERY position and the "." and "Answer" token positions which immediately follow QUERY.

**Fact-processing heads.** The fact-processing heads $\{(24, 5), (25, 7), (26, 0), (26, 12)\}$'s attention patterns at the ":" position tend to place larger weight on the correct fact for the answer, similar to the fact-processing heads in Mistral-7B. An interesting difference does exist though: heads (24,5) and (25,7) also tend to place a nontrivial amount of weight on the QUERY and ":" token positions, indicating that these heads are relying on some form of mixture of information present at those positions for processing. While it is reasonable to hypothesize that these heads are likely relying on the queried-rule information present in the QUERY and ":" residual streams, we have not confirmed this hypothesis in our current experiments. We visualize the statistics of these heads in Figure 23.

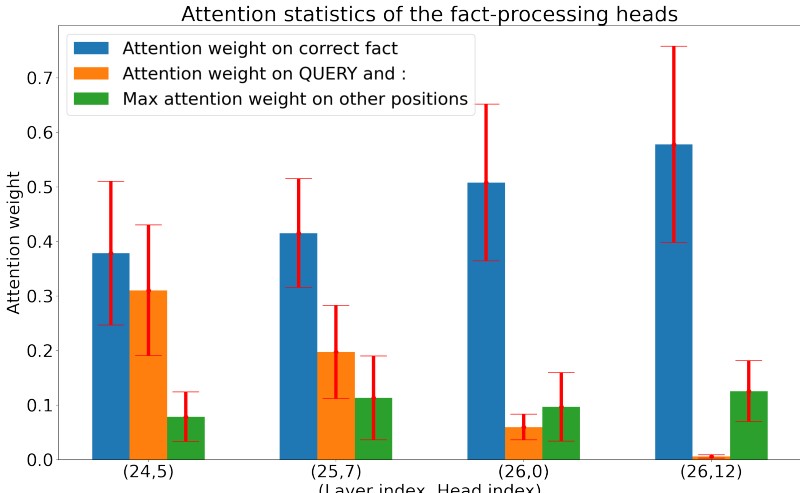

Figure 23: Average attention weights of the fact-processing heads in Gemma-2-9B, along with the standard deviations. The attention pattern is obtained at the ":" position. We record the attention weights at the correct fact, QUERY and ":" positions, and the maximum weight on any other position.

**Decision heads**. The decision heads $\{(28, 12), (30, 9)\}$'s attention pattern are obtained at the ":" position. They bear strong resemblance to those in Mistral-7B: they place significant attention on the correct answer token (in both the rules and facts sections, same as Mistral-7B's decision heads), and little attention weight anywhere else. This is shown in Figure 24.

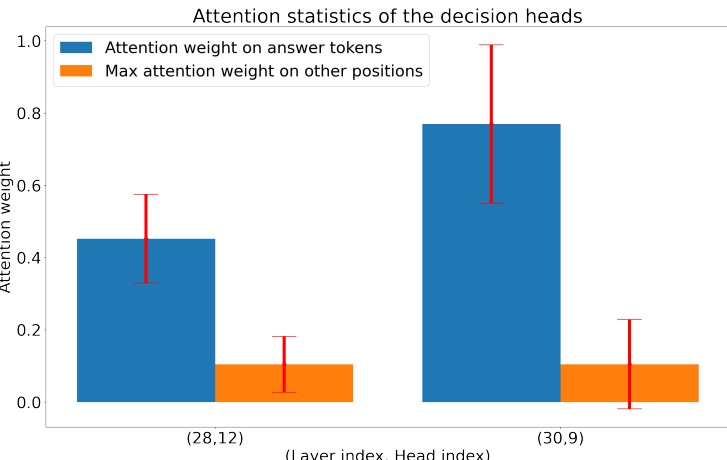

Figure 24: Average attention weights of the decision in Gemma-2-9B, along with the standard deviations. The attention pattern is obtained at the ":" position. We record the attention weights at the correct answer token positions.

### C.7.3 CIRCUIT VERIFICATION

We perform a sufficiency test of the attention-head circuit, following the same methodology as in the Mistral case, as discussed in §4.2 and C.4.

The circuit which we perform verification on is the union of the four attention head families, $\mathcal{C} = QRLH \cup QRMH \cup FPH \cup DH$, with

- $QRLH$ = Queried-Rule Locating Heads = $\{(19, 11), (21, 7), (22, 5), (23, 12)\}$;

- $QRMH$ = Queried-Rule Mover Heads = $\{(20, 7), (23, 6), (24, 15)\}$;
- $FPH$ = Fact-Processing Heads = $\{(24, 5), (25, 7), (26, 0), (26, 12)\}$;
- $DH$ = Decision Heads = $\{(28, 12), (30, 9)\}$.

*Remark.* The circuit verification is performed in a *coarser-grained* manner in this experiment, as we patch the output of the attention heads in $\mathcal{C}$ from the QUERY position to the ":" position, instead of clearly distinguishing the token positions which each head primarily focuses on.

| $\mathcal{C}^{\dagger}$ | $\mathcal{C}_{null}$ | $\mathcal{C}$ | $\mathcal{C} - QRLH$ | $\mathcal{C} - QRMH$ | $\mathcal{C} - FPH$ | $\mathcal{C} - DH$ |
|---|---|---|---|---|---|---|
| $\Delta^{\mathcal{C}^{\dagger}}_{orig \to alt}/\Delta_{alt}$ | -1.0 | **0.94** | -0.97 | -0.40 | 0.17 | -1.11 |

Table 2: $\Delta^{\mathcal{C}^{\dagger}}_{orig \to alt}/\Delta_{alt}$ for Gemma-2-9B, with different choices of $\mathcal{C}^{\dagger}$. $\mathcal{C}_{null}$ denotes the *empty* circuit, i.e. the case where no intervention is performed. We abbreviate the attention head families, for example, $DH$ = decision heads; $\mathcal{C} - DH$ = full circuit but with the decision heads removed.

We find that by patching all 13 attention heads in $\mathcal{C}$, $\Delta^{\mathcal{C}}_{orig \to alt}$ is about 94% of the "maximal" average logit difference $\Delta_{alt}$ on the altered samples. Moreover, removing any one of the four families of attention heads from $\mathcal{C}$ in the circuit interventions renders the "belief altering" effect of the intervention almost trivial.

*Remark.* We find it surprising that two LLMs (Mistral-7B and Gemma-2-9B) which are trained with different procedures and data ended up relying on attention-head circuits which bear strong resemblance to each other's. In the current literature, it is unclear how one can rigorously quantify the similarity of two nontrivial circuits inside different LLMs, however, this subsection does yield preliminary evidence that, the reasoning circuit we discover potentially has some degree of *universality* to it, and is likely an emergent trait of LLMs.

