# OpenReview forum: "How Transformers Solve Propositional Logic Problems: A Mechanistic Analysis"
_ICLR.cc/2025/Conference — Submitted to ICLR 2025_

### Official Review · Reviewer_qNQi · 2024-10-30

**Soundness:** 4
**Presentation:** 2
**Contribution:** 2
**Rating:** 3
**Confidence:** 4

**Summary:**

In this paper, the authors explore the mechanisms by which LLMs/transformers solve propositional logic problems. They work in two steps: first, they train a small transformer (3-layer 3-head) on a very specific propositional logic problems, to study in detail how transformers work, before considering a large pre-trained LLM (Mistral 7B) and trying to replicate their findings. The authors claim a very precise and human-like reasoning process (p4, section 3.2) for the transformer they trained.

**Strengths:**

Strengths:

S1. To understand how LLMs/transformers generalize symbolic problem is a very timely and interesting topic.

S2. The overall protocol to first test a small transformer then a larger LLM makes perfect sense.

S3. The first claim on a routing signal between logOp and Linear chains is convincing. (figure 2)

**Weaknesses:**

Weakness:

W1. The biggest shortcoming lies in the problem chosen. While Mistral 7B  generalizes from what it learnt, this is simply not the case of the transformers the authors trained, because the logical problems is overly simplistic: from what I understood, there are at most 10 cases considered, up to renaming of variables, and each of these cases is seen around 100.000 times by the transformer ( with different variable names). Hence, there is no generalization of reasoning, just a classification of which of the (at most 10) cases is witnesses (this is actually even partially backed-up by the own author experiments), and copy of the associated answer (among 10), with a correct instantiation of variable names. Far away from the advertised “solving of propositional problem” study in the title.


The at most 10 cases:

-the query symbol (A in the example) is in the linear chain, and the first symbol is true in Facts (A true).

-the query symbol (A in the example) is in the linear chain, and the first symbol is false in Facts (A false).


-the query symbol (A in the example) is in the LogOp, the operator is "and", and both first symbols are true (A true).

-the query symbol (A in the example) is in the LogOp, the operator is "and", and top input symbols is false in Fact (the other is true) (A false).

-the query symbol (A in the example) is in the LogOp, the operator is "and", and bottom input m symbol is false in Fact  (the other is true) (A false).

-the query symbol (A in the example) is in the LogOp, the operator is "and", and both input symbol are false in Fact  (A false).


-the query symbol (A in the example) is in the LogOp, the operator is "or", and both first symbols are true (A true).

-the query symbol (A in the example) is in the LogOp, the operator is "or", and top input symbols is true in Fact (the other is false) (A true).

-the query symbol (A in the example) is in the LogOp, the operator is "or", and bottom input symbol is true in Fact  (the other is false) (A true).

-the query symbol (A in the example) is in the LogOp, the operator is "or", and both input symbol are false in Fact  (A false).


Hence, the transformer does not learn to generalize, it just learns to check symbols at different places for equality, following at most 10 different patterns.

The fact that there is a strong routing as shown in Figure 2 is thus extremely unsurprising.

Actually, there seems to be more routing hidden in figure 2 (but not as clear as the logOp/Linear chain), e.g. whether "and"/"or" is the logical operator.


W2 (Edit: partially fixed) another important issue, the fact that there is no precise definition of the very logical problem considered (see questions below). One revealing case, the simple example of formula page 3 is incorrect:
Answer: …. "V implies E, E true" … , which is incorrect as "Facts: V False".
Also, “minimal” (proofs) should be defined and used correctly. There are at least different 2 reasonable definitions of minimality.

Some Questions:
Q1- Is there exactly one LogOp and exactly one Linear Chain. (It is what I guess after reading 6 pages, but this is not precisely explicitated, whereas this is a crucial information to understand the transformer being trained).
Q2 - Can some symbol be common between the LogOp and the Linear chain, in particular the final (query) symbol? If yes, then the 10+ sentences with "*the* linear case" and "*the* LogOp" case should be rewritten.

W3 Other claims on the transformers are much less convincing. For instance,
-(partially fixed) evidence 2 is a very weak evidence. This test can only tell whether the information you are looking for is present or not in the layer. So yes, the information is present in the third layer, but this is hardly surprising: it is obviously present in the input, and probably in every layer. And also probably in layers even for the linear case.
We would need to see such data (% of accuracy for classification) on other layers/cases, and also on other variables. (Relative) Comparison of the % of accuracy could give some indication on what is encoded in the last layer, absolute value (97%) is not conclusive.

-evidence 1c is not compatible with h_route being >0 or <0 depending on the case. Overall, the very precise protocol from the transformer seems a big stretch from the evidence.

W4 The case of Mistral 7B does not really confirm the findings on the trained transformers (which is not surprising, because the training process for the trained transformer cannot generalize, or only to choose variable names to output, not to solve the logical problem).
Question Q3: The logical formula checked for Mistal 7B is actually even restricted to shorter chains (lenght 2) than for the transformer (size 3). Why?


To sum up, it is an interesting problem, but the realisation is pretty weak and cannot be used to conclude much about how LLMs solve a logical problem. Hence, I recommend to reject the paper.

**Questions:**

Q1- Is there exactly one LogOp and exactly one Linear Chain. (It is what I guess after reading 6 pages, but this is not precisely explicitated, whereas this is a crucial information to understand the transformer being trained).
=> answer is yes.

Q2 - Can some symbol be common between the LogOp and the Linear chain, in particular the final (query) symbol? If yes, then the 10+ sentences with "*the* linear case" and "*the* LogOp" case should be rewritten.
=> answer is no.

Q3 - Why is Mistral 7B only queried with length 2 queries?

---

> ### Author Response · Authors · 2024-11-19
> **Response to Reviewer qNQi, Part 1**
>
> We appreciate your constructive feedback. Our response to your questions and comments is as follows.
>
> **Position and contribution of this work**. Before addressing your specific comments, we wish to clarify the goal of our work, and its position in the literature. First, as discussed in Section 1.1 of our paper, our paper belongs to the field of mechanistic interpretability, particularly the area of _circuit analysis_ of large language models. This area is new, and most existing studies have been performed on _highly simplified problems_ that grant accessibility to fine-grained analysis of information flow and processing inside the model, at the level of attention heads, MLPs, etc. [1-3]. For example, the pioneering circuit analysis work [1] studies the attention circuit in GPT2-small on a simple task of identifying the “missing variable” in a conjunctive sentence, where the prompt templates follow simple forms such as “When Mary _and_ John went to the store, John gave a drink to ”: the answer is “Mary”, the “missing variable” from the conjunctive sentence.
>
> The primary goal of our work is to _initiate_ the study of reasoning circuits in transformer-based language models, especially in pretrained LLMs. We wish to place special emphasis on our analysis of how Mistral-7B solves the reasoning problem, as it is, to the best of our knowledge, the first to characterize the critical components of the circuit employed by an LLM _in the wild_ for solving a nontrivial logical reasoning problem (involving distracting clauses) that requires a correct execution of the full reasoning path “Query→Relevant rule(s) →Relevant fact(s) →Decision'' fully in context. The reasoning problem, while simple, captures some of the core aspects of broader classes of reasoning problems in practice. Moreover, we believe that the techniques we developed for examining the necessity and sufficiency of the pretrained LLM's reasoning circuit is of concrete value to future research in _mechanistic interpretability_. _Therefore, we kindly ask you to consider the whole of the paper, especially our circuit analysis of Mistral-7B, in your assessment of the work._
>
> **Weaknesses**.
>
> **W1**, **W4** (partial) & **Q3**. “The biggest shortcoming lies in the problem chosen", “at most 10 cases considered, up to renaming of variables”, “no generalization of reasoning, just a classification of which of the (at most 10) cases is witnesses”, “The logical formula checked for Mistal 7B is actually even restricted to shorter chains (length 2)”, “Why is Mistral 7B only queried with length 2 queries?”.
>
> First, we hope our clarification on the position of this paper and its relation to existing works helps elucidate our choice of the logical reasoning problem despite its simplicity: the data model needs to be complex enough so that the problems are not solved through simple shortcuts (skipping steps in the “Query→Relevant rule(s) →Relevant fact(s) →Decision” chain we discussed above), and simple enough to facilitate interpretation and intervention studies — this simplicity is especially vital for intervention studies of LLMs in the wild (Mistral-7B in our case). Moreover, since the reasoning problem still requires the model to perform several actions that are essential to more complex reasoning problems, the analysis techniques and insights we developed can serve as the foundation for understanding not only how the LLM solves more complex reasoning problems, but also why it sometimes fails on those problems, by identifying anomalous information flow and processing in the LLM in those settings.
>
> Furthermore, while the reasoning problem is indeed simple, we respectfully disagree on the claim that the problem is simple due to “at most 10 cases considered, up to renaming of variables”. The language model only observes the text-level information, it never sees the underlying causal structure of the problem: with 80 proposition variables the problem can choose from, the number of distinct combinations of rules that can possibly appear at the text level is at least on the order of $10^{10}$ (even disregarding the possible combinations of truth value assignments and query choice), and the model has to employ a consistent strategy (such as the one chosen by the 3-layer 3-head model explained in Section 3) to avoid mistakes in the test samples. In general, we believe a "post hoc" characterization of a learning problem by ignoring the complexity of the raw input distribution, and only relying on certain aspects of the “complexity” of an already discovered solution can lead to severe under-estimation of the learning problem’s complexity.

---

> ### Author Response · Authors · 2024-11-19
> **Response to Reviewer qNQi, Part 2**
>
> **W2**. “no precise definition of the very logical problem considered”, “ "V implies E, E true" … , which is incorrect as ‘Facts: V False’”, “ “minimal” (proofs) should be defined and used correctly”
>
> We apologize for the typos in Section 2’s example. We have clarified the problem format and “minimal” proofs in our paper revision, particularly in Section 2, Appendix A and B.1.
>
> Now, as stated in Section 2 of the paper, we require the answer to a problem instance to be structured in the format of listing out the relevant facts first, followed by invoking the relevant rules in conjunction with a correct statement of the truth value of the “conclusion” variable of the invoked rule; irrelevant rules and facts must not be invoked. Therefore, in our example in Section 2, the correct “minimal” proof is:
> K TRUE. K implies D; D TRUE. D or E implies A; A TRUE.
>
> A slight subtlety lies in the cases of both starting variables for an OR chain being TRUE, or both starting variables for an AND chain being FALSE. In the former case, we count the reasoner’s answer correct as long as it only lists one of the TRUE facts and properly invokes the rules and determines the resulting truth value; similarly for the latter case (for an AND chain, the final answer will be UNDETERMINED in this case). These cases are clarified in the new Appendix A of the paper.
>
> **W3a**. “evidence 2 is a very weak evidence”, “the information is present in the third layer”, “also probably in layers even for the linear case …”
>
> Let us first ensure that we are on the same page about evidence 2. The purpose of the linear probing experiment in Evidence 2 is to verify that the answer for the linear chain is indeed linearly encoded in the layer-2 residual stream at the QUERY position, _when the linear chain is queried in the problem instance_. The reason we are performing this experiment is because of Evidence 1b (Lines 227-232): whenever the linear chain is queried, we find that the attention weights in the layer-3 attention heads at the ANSWER position are all almost 100% focused on the QUERY position, in other words, QUERY is the only _source_ position which layer-3 attention is moving to the target ANSWER position. This naturally leads to the question of whether the layer-2 residual stream at this position already has the answer for the linear chain: we are just doing a confirmatory experiment in Evidence 2 that the answer is indeed present there, and is in fact linearly encoded (due to the 97% test accuracy of the linear predictor, trained on 5k samples).
>
> For further contrasting evidence, we performed an additional experiment, by training a linear classifier on exactly the same task as in Evidence 2 (also with 5k training samples), but to predict the correct answer token of the LogOp chain. As expected, the classifier has a low test accuracy of approximately 27% when tested on 10k samples, and exhibits severe overfitting, with the training accuracy around 94%.
>
> **W3b**. “evidence 1c is not compatible with h_route being >0 or <0 depending on the case”
>
> Evidence 1c is exactly there to show that the model’s behavior, particularly behavior of the third layer and the model’s output, is causally dependent on (the “sign” of) this $h_{route}$ direction. We reiterate the main points of Evidence 1c here. We show evidence that if QUERY is for the linear chain, when we subtract the estimated $h_{route}$ from layer-2 attention block’s output at the QUERY position (so we push the correlation of this output embedding with $h_{route}$ towards the negative direction), the third layer attention behaves in almost the same way as it does when QUERY is for the LogOp chain, that is, it not only suppresses attention at the QUERY position, but also shift attention to the Rules and Facts section (in fact in this case, the model consistently outputs the correct answer token for the LogOp chain after the intervention). On the other hand, if QUERY is for the LogOp chain, adding the estimated $h_{route}$ causes the third layer attention to focus almost 100% on the QUERY position, making the third layer behave as if QUERY is for the linear chain.

---

> > ### Comment · Reviewer_qNQi · 2024-11-21
> >
> > Thanks the authors for their precision.
> >
> > W3a. I overlooked the fact that the classifier is linear, not a general DNN.
> >
> > You agree that a generic DNN could find this information in probably all the layers for all cases?
> >
> > Thanks for the additional test. However, it does not help much:
> > is the output of the LINEAR chain still encoded linearly when the query is on the LogOp chain?
> > This could very well be the case with high accuracy. There are enough bits of information to keep that information no matter what.
> >
> > The fact that the answer of the LogOp chain is not linearly encoded does not really matters, it is encoded in another way.
> >
> > Question: What is so important in the fact that the encoding is linear?
> >
> > For W3b: "In this case, however, the model does not output the correct starting node for the linear chain on more than 90% of the test samples", this conflicts with the >0 / <0 .

---

> > > ### Author Response · Authors · 2024-11-22
> > > **Response to Reviewer qNQi's comments, Part 1**
> > >
> > > **C1**. “ generic DNN could find this information in probably all the layers for all cases”, “There are enough bits of information to keep that information no matter what”, “The fact that the answer of the LogOp chain is not linearly encoded does not really matters, it is encoded in another way.”, “What is so important in the fact that the encoding is linear?”
> > >
> > > Please refer to our paper’s sections 3.2.1 and 3.2.2 and our response to W3a and W3b above for a detailed answer to your comments. We will briefly reiterate some important points here. Recall that layer 3 places almost 100% attention at the QUERY token position position whenever the linear chain is queried. This means that only the embedding at the QUERY position at layer 2 gets passed to the ANSWER position at layer 3. Now, the rest of the transformer’s operations on this embedding for producing the output are almost all _linear_: the embedding will pass through the value matrix (_linear_), the projection matrix (_linear_), the layer normalization (the only slightly nonlinear part), and finally through the unembedding layer (_linear_) to produce the answer token logits. This raises the obvious question of whether it is already possible to linearly decode the answer from the layer-2 residual stream at the QUERY position. We confirm this hypothesis in Evidence 2. For further contrasting evidence, we find that it is not possible to linearly decode the LogOp chain’s answer at this position, which we already discussed in detail in our rebuttal above.
> > >
> > > **C2**. "In this case, however, the model does not output the correct starting node for the linear chain on more than 90% of the test samples", this conflicts with the >0 / <0 .
> > >
> > > We believe that this misunderstanding likely springs from misinterpretation of the evidence. _The function of the routing signal in layer 2 is in determining the **action of the third layer attention block**_. Here is the natural behavior of layer-3 attention: (1) when the linear chain is queried, the third layer concentrates almost 100% of the attention on the QUERY position; (2) when the LogOp chain is queried, almost 0% attention is placed on the QUERY position, and instead attention is shifted to the Rules and Facts section.
> > >
> > > Now, our intervention experiment shows that we can exactly flip the behavior of the third layer by subtracting (adding) the estimated $h_{route}$ vector when the linear (LogOp) chain is queried. This yields evidence that the sign of the $h_{route}$ signal indeed determines the internal information flow of the small transformer (in the third layer). This is the core conclusion of Evidence 1c.

---

> > > > ### Comment · Reviewer_qNQi · 2024-11-23
> > > >
> > > > >However, it does not help much: is the output of the LINEAR chain still encoded linearly when the query is on the LogOp chain?
> > > >
> > > > can you please answer this?

---

> > > > > ### Author Response · Authors · 2024-11-25
> > > > > **Response to Reviewer qNQi's comment**
> > > > >
> > > > > >However, it does not help much: is the output of the LINEAR chain still encoded linearly when the query is on the LogOp chain?
> > > > >
> > > > > > can you please answer this?
> > > > >
> > > > > **Setting**. Due to the causal nature of the reasoning problem (it is only possible to know the answer at or after the QUERY token position), and the causal nature of the decoder-only transformer, we trained a set of linear classifiers on all token positions at or after the QUERY token and up to the ANSWER token, and on all layers of the residual stream of the transformer. We follow the same procedure as before — except in this set of experiments, for contrasting evidence, QUERY is for the LopOp chain, while the classifier is trained to predict the answer of the Linear chain.
> > > > >
> > > > > **Results**. The maximum test accuracy of the linear classifiers across all aforementioned token positions and layer indices is only 32.7%. That is, the answer of the Linear chain is _not_ linearly encoded in the model when QUERY is for the LopOp chain.

---

> > > > > > ### Comment · Reviewer_qNQi · 2024-11-27
> > > > > >
> > > > > > Thanks for the experiment. This is something you need to provide in the appendix to make the evidence more solid.
> > > > > >
> > > > > > by the way why "all layer indices"? You should keep the test as close to the test for Linear chain as possible, ie check the first layer only (?).

---

> ### Author Response · Authors · 2024-11-19
> **Response to Reviewer qNQi, Part 3**
>
> **W4**. “The case of Mistral 7B does not really confirm the findings on the trained transformers”
>
> We wish to clarify that the small transformer and Mistral-7B experiments are independent results. We did not state that the Mistral results should align exactly with the small transformer results: our intention is to present results on the two fronts for the sake of completeness in studying how transformers solve the simple logical reasoning problem.
>
> Furthermore, at least in the current stage, we do not believe that one should expect the reasoning circuit inside a general-purpose LLM in the wild to be the same as a special-purpose small transformer trained on the logic problems (it is not even clear how one would quantify similarity between two circuits as of now). Strong hints of disagreement of reasoning mechanisms between special- and general-purpose language models had already been observed in multiple places in the literature. For instance, [4,5] show that general-purpose LLMs (GPT-4-turbo, Gemini Pro, etc.) tend to be less robust to rule-order shuffling and distracting clauses than special-purpose small transformers trained purely on the propositional logic datasets. A more recent example is [6], which shows that small transformers trained only on GSM-like problems are more robust and generalizing on (more complex classes of) these problems than the general-purpose SOTA LLMs. _Overall, we believe that the reasoning mechanisms of both special-purpose small transformers and general-purpose LLMs are worth studying in their own right._
>
> **Q1**. “Is there exactly one LogOp and exactly one Linear Chain?”
>
> Yes, there is exactly one LopOp chain and exactly one linear chain.
>
> **Q2**. “Can some symbol be common between the LogOp and the Linear chain, in particular the final (query) symbol?”
>
> The two chains cannot share any proposition variable.
>
> **References**.
>
> 1. Kevin Ro Wang, Alexandre Variengien, Arthur Conmy, Buck Shlegeris, and Jacob Steinhardt.
> Interpretability in the wild: a circuit for indirect object identification in GPT-2 small. In
> ICLR, 2023.
>
> 2. Michael Hanna, Ollie Liu, and Alexandre Variengien. How does gpt-2 compute greater-than? interpreting mathematical abilities in a pre-trained language model. In NeurIPS, 2023.
>
> 3. Neel Nanda, Lawrence Chan, Tom Lieberum, Jess Smith, and Jacob Steinhardt (2023). “Progress measures for grokking via mechanistic interpretability”. In ICLR, 2023.
>
> 4. Xinyun Chen, Ryan A Chi, Xuezhi Wang, and Denny Zhou. 2024. Premise order matters in reasoning with large language models. arXiv preprint arXiv:2402.08939.
>
> 5. Honghua Zhang, Liunian Harold Li, Tao Meng, Kai-Wei Chang, and Guy Van Den Broeck. On the paradox of learning to reason from data. In Proceedings of the Thirty-Second International Joint Conference on Artificial Intelligence, IJCAI ’23, 2023.
>
> 6. Tian Ye, Zicheng Xu, Yuanzhi Li, and Zeyuan Allen-Zhu. Physics of language models: Part 2.1, grade-school math and the hidden reasoning process. arXiv preprint arXiv:2407.20311, 2024.

---

> > ### Comment · Reviewer_qNQi · 2024-11-21
> >
> > answers to Q1 and Q2 confirm that we are looking at 10 patterns up to renaming.
> >
> > Thus, the smaller transformer only learns to identify the pattern (very easy - a linear classifier can do that), and apply the answer with renaming. That's not solving a propositional logic problem.
> >
> > The analysis of the Transformer and of Mistral 7B are thus perfectly orthogonal.
> >
> > If "the small transformer and Mistral-7B experiments are independent results", then why write the paper with the two results?
> >
> > Looking at the difference of both is not interesting. On the other hand, what would be interesting is to generalize the findings of the small Transformer to the large LLM.

---

> ### Author Response · Authors · 2024-11-22
> **Response to Review qNQi's comments, Part 2**
>
> **C3**. “the smaller transformer only learns to identify the pattern (**very easy - a linear classifier can do that**), and apply the answer with renaming. That's not solving a propositional logic problem”
>
> We argue that the claim that a linear classifier can solve this problem is demonstrably **false**. As shown in Figure 6 in the paper (on page 17), the transformer models with fewer layers than 3 have nontrivial error rates — **these transformer models are more powerful than a linear classifier**,  and still cannot perfectly solve this problem.
>
> We would also like to emphasize that mechanistic analysis is not about generalization capabilities of models but rather the goal is to understand the models. We already know that LLMs are incredibly powerful and do generalize in mysterious ways. Our focus here is to shed some light on this mystery. In particular, we are not claiming our paper as evidence that transformers solve propositional logic problems in their entirety.
>
>
> **C4a**. “If ‘the small transformer and Mistral-7B experiments are independent results’, then why write the paper with the two results?”.
>
> The unifying theme of our work is a mechanistic analysis of how transformers plan and reason to solve a propositional logic problem. As we just argued, **this problem is too challenging for one- or two -layer transformers.** Hence, we study the precise mechanisms that larger transformers employ to solve the problem. We chose two natural settings for larger models: (a) a minimum model, the 3-layer transformer, and (b) a pre-trained general purpose LLM, Mistral 7B. Then, we discuss in detail the underlying transformer components (e.g., attention heads) that enable (a) and (b) to plan and solve the logic problems. Compared to prior work, _the unification from studying a single problem type is a novel contribution, as are the nuanced internal circuits that we discover._
>
> Furthermore, we argue that the difference between reasoning circuits in small transformers and pretrained LLM (Mistral 7B) is worthy of interest. In most of the previous circuit analysis works, each work only identifies one algorithm which a single transformer relies on to solve the problem of interest [1-3]. In contrast, our work demonstrates that transformers are capable of solving the same propositional logic problem with multiple different, yet correct algorithms.
>
> **C4b**. “Looking at the difference of both is not interesting. On the other hand, what would be interesting is to generalize the findings of the small Transformer to the large LLM.”
>
> This is a highly subjective judgement of the worth of a scientifically valid investigation, and is an unreasonable metric to evaluate the value of our work upon. Mechanistic analysis is not directly about generalization, as the goal is to _understand_ the models, not to _design_ new models.
>
> **References**.
>
> 1. Kevin Ro Wang, Alexandre Variengien, Arthur Conmy, Buck Shlegeris, and Jacob Steinhardt.
> Interpretability in the wild: a circuit for indirect object identification in GPT-2 small. In
> ICLR, 2023.
>
> 2. Michael Hanna, Ollie Liu, and Alexandre Variengien. How does gpt-2 compute greater-than? interpreting mathematical abilities in a pre-trained language model. In NeurIPS, 2023.
>
> 3. Nanda, Neel, Lawrence Chan, Tom Lieberum, Jess Smith, and Jacob Steinhardt (2023). “Progress measures for grokking via mechanistic interpretability”. In ICLR, 2023.

---

> ### Comment · Reviewer_qNQi · 2024-11-23
>
> >>C3. “the smaller transformer only learns to identify the pattern (very easy - a linear classifier can do that), and apply the answer with renaming. That's not solving a propositional logic problem”
>
>
> >We argue that the claim that a linear classifier can solve this problem is demonstrably false. As shown in Figure 6 in the paper (on page 17), the transformer models with fewer layers than 3 have nontrivial error rates.
>
> You misunderstood. I was not claiming that writting the correct minimal answer  could be done using a linear classifier (and indeed, figure 6 shows it is not possible), I was only claiming that
>
> "identify the pattern" (check where the parenthesis is placed, just after "identify the pattern", not at the end of the sentence).
>
> could be done extremely easily. This is a very simple pattern matching of only 10 cases, where one only needs to match the names of 2 or 3 fixed variables.
>
>
> >"the goal is to understand the model"
>
>
> Very well, but your model is only trained to pattern match 10 cases (very easy, see above), then write a fixed answer. The only slightly complicated task is to rename those variables in the fixed answer to the one appearing in the query.  That s where the heads are useful, to follow the chain / LogOp to pick the right variable names.
>
>
> If you want to write a paper on mechanistic analysis of pattern matching and renaming, then you should change the title of the paper. But then it would be much less interesting.

---

> ### Author Response · Authors · 2024-11-25
> **Response to Reviewer qNQi**
>
> We want to first sincerely thank you for the continued discussion and speedy replies. As the discussion period is coming to an end, we want to prioritize emphasizing and clarifying the important points of our rebuttal, _a good portion of which have not received enough attention in this discussion_.
>
> 1. **Contribution**. In addition to our analysis of how the special-purpose small transformer solves the reasoning problem, _we place special emphasis on our analysis of the reasoning circuit of Mistral-7B, which occupies a much larger portion of the paper and our overall contribution_. This analysis is, to the best of our knowledge, **the first to characterize the critical components of the circuit employed by a pretrained LLM _in the wild_ for solving a nontrivial logical reasoning problem (involving distracting clauses) that requires a correct execution of the reasoning path “Query→Relevant rule(s) →Relevant fact(s) →Decision'' fully in context**. We believe that the techniques we developed for examining the _necessity_ and _sufficiency_ of the pretrained LLM's reasoning circuit is of concrete value to future research in _mechanistic interpretability_.
> - **Relation to existing work**. Circuit analysis of pretrained LLMs in the wild requires significantly simplifying the problems of interest to facilitate precise interpretation and intervention studies. Consider for example the work [1], the most related circuit analysis work to ours, which studies how _GPT2-small_ solves simple problems with prompts of the form “When Mary and John went to the store, John gave a drink to ” [Answer is “Mary”], involving only a _variable-relation_ statement. In contrast, we consider how Mistral-7B (a much larger and more modern LLM) solves a nontrivial full-reasoning problem, demanding the LLM to properly process the _variable relations, value assignments and query_, and _correctly ignore the distracting clauses_. Moreover, our necessity- and sufficiency-based studies of the LLM’s reasoning circuit is also a first in the literature to the best of our knowledge.
>
> 2. **Special-purpose vs. general-purpose models**. By the current literature, **it is perfectly reasonable to expect the reasoning circuit inside a general-purpose LLM in the wild to _differ_ significantly from that in a special-purpose small transformer trained on the logic problems**. For instance, [2,3] show that general-purpose LLMs (GPT-4, Gemini, etc.) tend to be much less robust to rule-order shuffling and distracting clauses than special-purpose small transformers trained purely on the propositional logic datasets. Another recent example is [4], which shows that small transformers trained only on GSM-like problems are much more robust and generalizing on (more complex classes of) these problems than the general-purpose SOTA LLMs. Overall, we believe that the hidden mechanisms of both special- and general-purpose transformers are worth studying in their own right. _Moreover, opinions such as “Looking at the difference of both is not interesting. On the other hand, what would be interesting is to generalize the findings of the small Transformer to the large LLM” is not supported well by current evidence._
>
>
>
> **References**.
>
> 1. Kevin Ro Wang, Alexandre Variengien, Arthur Conmy, Buck Shlegeris, and Jacob Steinhardt.
> Interpretability in the wild: a circuit for indirect object identification in GPT-2 small. In ICLR, 2023.
>
> 2. Xinyun Chen, Ryan A Chi, Xuezhi Wang, and Denny Zhou. 2024. Premise order matters in reasoning with large language models. In ICML, 2024.
>
> 3. Honghua Zhang, Liunian Harold Li, Tao Meng, Kai-Wei Chang, and Guy Van Den Broeck. On the paradox of learning to reason from data. In IJCAI, 2023.
>
> 4. Tian Ye, Zicheng Xu, Yuanzhi Li, and Zeyuan Allen-Zhu. Physics of language models: Part 2.1, grade-school math and the hidden reasoning process. arXiv preprint arXiv:2407.20311, 2024.

---

> ### Comment · Reviewer_qNQi · 2024-11-27
>
> I'd say there is fundamental problems with the analysis of the small trained Transformers, mainly because of the problem considered.
>
> If the authors truely believe that:
>
> > Mistral-7B occupies a much larger portion of the paper and our overall contribution.
>
> Then I would suggest to just remove the problematic part, and focus on Mistral-7B. There is enough material in the appendix to make a paper.
>
> As this is not what is currently implemented in the PDF, I cannot recommand acceptance.
>
> By the way, I do not think i saw an answer to my question: why Mistral 7B is only queried with even simpler chains than the Transformer (size 2 vs size 3).

---

> > ### Comment · Reviewer_H2ZQ · 2024-11-27
> >
> > I am writing in support of the authors.
> >
> > I find their analysis techniques of both small and large models interesting. In general, it is quite hard to investigate large models and establish definitive statements. Thus, I am okay with interesting findings, even if the setup and conclusions might be less than ideal. I will personally be looking to apply their techniques and related analyses in my work.
> >
> > I would also much rather see papers like this rather than the flood of prompt engineering-based reasoning work that's currently under submission. While there are areas for improvement, I feel that this paper meaningfully advances machine learning research, particularly in how to better study LLMs.

---

> ### Author Response · Authors · 2024-11-29
>
> Thank you for the replies. As the extended discussion period is also coming to an end, we will make a somewhat more general counterpoint on the finer claims below:
>
> > "there is fundamental problems with the analysis of the small trained Transformers", "the smaller transformer only learns to identify the pattern (very easy - a linear classifier can do that), and apply the answer with renaming", etc.
>
> While the logic problem is intentionally simple in nature (as explained in the responses of ours above), we argue that it is incorrect to claim that a reasoner (e.g. a trained transformer) simply performs "pattern matching" followed by "variable renaming" to solve the problem.
>
> Given a problem instance, the first answer token requires identifying the _correct fact_ for the _minimal_ proof. The information which the reasoner sees, however, contains much ambiguity within 0 step of reasoning: there are three facts presented, which one is the correct one to write down? It _has to_ resolve this ambiguity as follows: (1) rely on the QUERY token in this problem instance, search in the Rules, to resolve the ambiguity of which tokens are the true "premise" tokens corresponding to the "premise" nodes of the queried chain, and which LopOp (AND vs. OR vs. identity) is present, (2) rely on the identified "premise" tokens to resolve the ambiguity of which facts are truly relevant. Now the reasoner reaches the correct answer, and the "pattern" of this problem has been identified. Therefore, _once the reasoner has figured out which one out of all the possible "patterns" is present in this problem instance, the reasoner has also simultaneously reached the token-level answer_.
>
> In other words, "pattern recognition" and "producing token-level answer" are entangled in the process of answering the problem instance. This is what generally happens with nontrivial deductive reasoning — _the reasoner cannot know, by a superficial examination of the context, which one out of all possible "templates" this problem instance belongs to, instead, it must thoroughly examine the context, and employ the instance-dependent information to resolve ambiguities in the possible answers, through possibly multiple steps of intermediate "ambiguity resolutions"_.
>
> [To ensure that we are on the same page, we reiterate that for each problem instance, the rules and facts sentences are presented in a random order in the respective sections of the context. This prevents the model from adopting position-based shortcuts in solving the problem.]

---

> > ### Comment · Reviewer_MVJS · 2024-11-29
> >
> > I agree with the comments made by Reviewer H2ZQ. This paper presents an interesting experiment of analysis and show cases that transformers in general and 2 well-known LLMs are able to perform reasoning to some extent via learned circuitry.
> >
> > I agree with your statements that the problem setting could be improved but I believe that the paper already makes significant contributions and analyzes large models as well. This could pave the way for future work to tackle harder problems such as more complicated propositional formulae, first-order logic, the lambda-calculus etc.
> >
> > IMO this paper provides a good analysis that tries to derive conclusions w.r.t. the architecture. In contrast, most approaches in the literature just conduct experiments at the prompt level to solve problems and conclude whether LLMs can/cannot reason.

---

### Official Review · Reviewer_qubH · 2024-10-31

**Soundness:** 2
**Presentation:** 2
**Contribution:** 1
**Rating:** 3
**Confidence:** 4

**Summary:**

This paper investigates the features that allow transformers and LLMs to reason about problems formulated in propositional logic. It also tries to derive some general lessons from these findings.

**Strengths:**

The authors manage to put their research in context and describe to some detail the current state of the art.

**Weaknesses:**

1) The authors are able to identify some feature of small transformers and Mistral-7B, an LLMs, which seem to facilitate learning and solving problems in propositional logic. But the actual impact of these features is not exactly clear.
Also, at the end of the introduction the authors discuss a set of assumptions they make, in order to keep the problem at hand tractable, which however impact on the generality and significance of their findings.

2) The type of problems as described in Sec. 2 is rather specific and restricted. So, the value of the contribution is somewhat limited.

3) The significance of the results obtained in the paper is not clear. I'd say that the methodology itself is not clear: what insight are to be obtained by analysing the structure of transformers? Moreover, the usefulness and applicability of such results is not clear either.
The evidence discovered seems to be mainly anecdoctic, pointing to very specific feature of the architectures, whose meaning or interpretation is quite obscure.

**Questions:**

1) The authors say that "these models exhibit deep language understanding and generation skills". It is not clear in which sense the understand of language that LLMs have is deep, as it is mainly next token prediction.

2) In the answer on p. 3, it appears that "V implies E; E TRUE". It is not clear to me why E should be true. Also, it is not required that E is true to derive A.

3) It is not clear what the authors mean when they say that "the transformer mentally process the context and plan its answer". In what sense can we assign a mind to transformers?

---

> ### Author Response · Authors · 2024-11-19
> **Response to Reviewer qubH, Part 1**
>
> Thank you for the constructive feedback. Here is our response to your questions and comments.
>
> **Position and contribution of this work**. Before addressing your specific comments, we wish to clarify the goal of our work, and its position in the literature. First, as discussed in Section 1.1 of our paper, our paper belongs to the field of mechanistic interpretability, particularly the area of _circuit analysis_ of large language models. This area is new, and most existing studies have been performed on _highly simplified problems_ that grant accessibility to fine-grained analysis of information flow and processing inside the model, at the level of attention heads, MLPs, etc. [1-3]. For example, the pioneering circuit analysis work [1] studies the attention circuit in GPT2-small on a simple task of identifying the “missing variable” in a conjunctive sentence, where the prompt templates follow simple forms such as “When Mary _and_ John went to the store, John gave a drink to ”: the answer is “Mary”, the “missing variable” from the conjunctive sentence.
>
> The primary goal of our work is to _initiate_ the study of reasoning circuits in transformer-based language models, especially in pretrained LLMs. We wish to place special emphasis on our analysis of how Mistral-7B solves the reasoning problem, as it is, to the best of our knowledge, the first to characterize the critical components of the circuit employed by an LLM _in the wild_ for solving a nontrivial logical reasoning problem (involving distracting clauses) that requires a correct execution of the full reasoning path “Query→Relevant rule(s) →Relevant fact(s) →Decision'' fully in context. The reasoning problem, while simple, captures some of the core aspects of broader classes of reasoning problems in practice. Moreover, we believe that the techniques we developed for examining the necessity and sufficiency of the pretrained LLM's reasoning circuit is of concrete value to future research in _mechanistic interpretability_.
>
> **Weaknesses**:
>
> **W1a**. “The authors are able to identify some feature …, But the actual impact of these features is not exactly clear”
>
> In addition to the probing results, these “features” have clear causal influence on the LLMs we examined. For small transformers, in Lines 236 - 350, we test the causal effects of the routing signal, and perform activation patching for demonstrating the causal effect of the “partial answer” in the layer-2 embeddings for providing an answer for the LopOp chain. For Mistral-7B, the whole section relies upon causal mediation analysis, which demonstrates the necessity of the reasoning circuit components we find by measuring their causal influence on the model’s correct output. We explain the mediation analysis techniques in Section 4.2, and show the results of the analysis in Section 4.3. We find that there is a surprisingly small set of attention heads (with strong causal effects on the model’s answer) for processing different sections of the problem context, as shown in Figures 4, 5 and 6.
>
>
> **W1b** & **W2**. “... assumptions they make … impact on the generality and significance of their findings.”, “The type of problems as described in Sec. 2 is rather specific and restricted. So, the value of the contribution is somewhat limited.”
>
> In addition to our comment on the position and contribution of this paper we made in the beginning, we wish to emphasize that current works in mechanistic interpretability, especially those in the area of circuit analysis, focus primarily on highly simplified problems to ensure the tractability of the analysis. For instance, our work innovates on at least two fronts compared to the pioneering circuit analysis work of [1] described before, in the following sense: (1) they focus on a simple problem containing only a statement of variable relations (without any distracting clauses too), while we study the circuit of an LLM for solving a full reasoning problem involving variable relations, value assignments and query, and include distracting clauses for more difficulty, and (2) they study GPT2-small, while we study Mistral-7B, a much larger and more modern pretrained LLM.
>
> Moreover, basic propositional logic is a fundamental task and building block of LLM reasoning. Understanding them is necessary towards broadly understanding how LLMs reason.

---

> ### Author Response · Authors · 2024-11-19
> **Response to Reviewer qubH, Part 2**
>
> **W3**. “what insight are to be obtained by analyzing the structure of transformers?”
>
> In general, by understanding underlying mechanisms of how the LLM performs inference, we can better predict out-of-distribution behavior of the LLM, better debug the models (identify and fix model errors), and understand emergent behavior. More specifically, the techniques and insights we developed in the simpler settings can serve as the foundation for understanding not only how the LLM solves more complex reasoning problems, but also why it sometimes fails on those problems, by identifying anomalous information flow and processing in the LLM in those settings. We also wish to point out that, it is already surprising that there even is a reasoning circuit (consisting of a very small set of attention heads) in an LLM in the wild (Mistral-7B) used for solving the reasoning problem; without our causal intervention experiments, it is entirely feasible to believe that the computations inside the LLM are highly _diffuse_ when they solve reasoning problems.
>
>
> **Questions**.
>
> **Q1**. “It is not clear in which sense the understand of language that LLMs have is deep, as it is mainly next token prediction”
>
> We respectfully disagree with the reviewer’s assessment of LLMs’ language understanding abilities and the justification being “mainly next token prediction”. Current (general-purpose) decoder-only LLMs do perform well on language understanding benchmarks, see for example benchmark results on MMLU, with larger LLMs such as GPT-4 attaining a 86.4% score, and smaller LLMs such as Mistral-7B attaining a 60.1% score, and related benchmark results in references [4-6].
>
> Moreover, on specific algorithmic tasks, transformers trained via next-token prediction had been observed to possess sophisticated understanding of the reasoning problems. For example, simply by training transformers via next-token prediction on human-produced chess games, [7] finds that it is possible to linearly decode board states and player skill level from the model’s hidden representations. Another recent example is [8], which shows that for transformers trained on GSM-styled problems, it is possible to linearly decode variable relations in the deeper layers of the model.
>
> Additionally, theory works such as [9] provide evidence that autoregressive next-word predictors (with CoT) are universal learners if trained properly.
>
>
> **Q2**. “In the answer on p. 3, it appears that "V implies E; E TRUE". It is not clear to me why E should be true. Also, it is not required that E is true to derive A.”
>
> We apologize for the typo. The correct answer should be: K TRUE. K implies D; D TRUE. D or E implies A; A TRUE. We will correct this in the paper revision.
>
>
> **Q3**. “It is not clear what the authors mean when they say that ‘the transformer mentally process the context and plan its answer’”.
>
> By “mentally” we mean “without chain of thought”.
>
> **References**.
>
> 1. Kevin Ro Wang, Alexandre Variengien, Arthur Conmy, Buck Shlegeris, and Jacob Steinhardt.
> Interpretability in the wild: a circuit for indirect object identification in GPT-2 small. In
> ICLR, 2023.
>
> 2. Michael Hanna, Ollie Liu, and Alexandre Variengien. How does gpt-2 compute greater-than? interpreting mathematical abilities in a pre-trained language model. In NeurIPS, 2023.
>
> 3. Nanda, Neel, Lawrence Chan, Tom Lieberum, Jess Smith, and Jacob Steinhardt (2023). “Progress measures for grokking via mechanistic interpretability”. In ICLR, 2023.
>
> 4. Josh Achiam, Steven Adler, Sandhini Agarwal, Lama Ahmad, Ilge Akkaya, Florencia Leoni Aleman, Diogo Almeida, Janko Altenschmidt, Sam Altman, Shyamal Anadkat, et al. 2023. Gpt-4 technical report. arXiv preprint arXiv:2303.08774, 2023.
>
> 5. Jiang, A. Q., Sablayrolles, A., Mensch, A., Bamford, C., Chaplot, D. S., de Las Casas, D., Bressand, F., Lengyel, G., Lample, G., Saulnier, L., Lavaud, L. R., Lachaux, M., Stock, P., Scao, T. L., Lavril, T., Wang, T., Lacroix, T., and Sayed, W. E. Mistral 7b. arXiv preprint arXiv:2310.06825, 2023.
>
> 6. Gemma Team, Morgane Riviere, Shreya Pathak, Pier Giuseppe Sessa, Cassidy Hardin, Surya Bhupatiraju, Leonard Hussenot, Thomas Mesnard, Bobak Shahriari, Alexandre Rame, et al. Gemma 2: Improving open language models at a practical size. arXiv preprint arXiv:2408.00118, 2024.
>
> 7. Adam Karvonen. Emergent World Models and Latent Variable Estimation in Chess-Playing Language Models. arXiv preprint arXiv:2403.15498, 2024.
>
> 8. Tian Ye, Zicheng Xu, Yuanzhi Li, and Zeyuan Allen-Zhu. Physics of language models: Part 2.1, grade-school math and the hidden reasoning process. arXiv preprint arXiv:2407.20311, 2024.
>
> 9. Malach, E. Auto-regressive next-token predictors are universal learners. In ICML, 2024.

---

> ### Author Response · Authors · 2024-11-22
> **Response to Reviewer qubH**
>
> Dear Reviewer,
>
> We hope that we have addressed your concerns through our paper revisions and the responses above. As the discussion period is coming to an end soon, please do not hesitate to share any remaining concerns you may have.
>
> Best,
> Authors of paper 13137

---

> > ### Comment · Reviewer_qubH · 2024-11-26
> >
> > I thank the authors for their lengthy explanations.
> >
> > Still. I have a couple of points which require further clarification.
> >
> > 1) The authors consider a rather specific problem, defined in the introduction. It is not clear how this allows to make conclusions about general propositional logic problems.
> >
> > 2) In the rebuttal the authors mention that their results might allow to debug models better. I'd like to see some empirical evidence of this statement. Do the authors have experimental results?
> >
> > Best

---

> ### Author Response · Authors · 2024-11-26
> **Response to Reviewer qubH**
>
> Thank you for your response! We will address your comments and questions below.
>
> > The authors consider a rather specific problem, defined in the introduction. It is not clear how this allows to make conclusions about general propositional logic problems.
>
> We choose a problem that _generalizes_ related works in the area of _circuit-based mechanistic interpretability_, and is reminiscent of problems studied in the recent _propositional-logic-based benchmark and analysis works_. The purpose of this work is not to study how LLMs solve fully general propositional logic problems, but to initiate the study of precisely localizing LLMs' reasoning circuits on tractable reasoning problems.
>
> We elaborate on our answer below.
>
> First, our choice of the propositional logic reasoning problems is heavily inspired by those studied in the recent propositional-logic-based benchmark and analysis papers such as [1-3]. [1] and [2] only contain propositional problems with _definite clauses_, and the _only_ logical operator which possibly appears in a rule is _conjunction_; [2] restricts the problems even further by only considering propositional problems with rules of the form $A \land B \implies C$ or $A \implies C$, with the answer always being True, and _only_ require proper applications of modus ponens (they are mainly checking proof accuracy in their studies). [3] is even simpler, as it only contains problems involving implications in its rules.
>
> Moreover, circuit analysis of pretrained LLMs in the wild requires significantly simplifying the problems of interest to facilitate precise interpretation and intervention studies. Consider for example the work [4], the most related circuit analysis work to ours, which studies how GPT2-small solves problems with prompts of the form “_When Mary and John went to the store, John gave a drink to_ ” [Answer is “Mary”], involving only a _variable-relation_ statement. In contrast, we consider a nontrivial full-reasoning problem, demanding the LLM to properly process the _variable relations, value assignments and query_, and _correctly ignore the distracting clauses_.
>
>
> >In the rebuttal the authors mention that their results might allow to debug models better. I'd like to see some empirical evidence of this statement. Do the authors have experimental results?
>
> At a high level, our work’s value, at least on the front of LLM debugging and alignment, is more on laying down the foundation: we show that it is possible to localize a _human-interpretable reasoning circuit_ inside an LLM (in the wild), and prove its necessity and sufficiency through delicately designed causal intervention experiments.
>
> To elaborate, one of the _ultimate_ goals of the field of (circuit-based) mechanistic interpretability is to use the results of circuit analysis to debug and align the LLMs. However, **we are at the very beginning of this pursuit, especially on the topic of logical reasoning**. This work is, to our knowledge, the first to show the _existence_ of reasoning circuits (and precisely localize them) inside a pretrained LLM in the wild for implementing a full “Question→Rules→Facts→Decision” reasoning chain, and that this circuit is so localized that the functions and composition of its components are _humanly interpretable_.
>
> This foundational understanding of the LLM’s reasoning circuit can serve many uses. For instance, the discovered circuit can be used as the baseline for contrasting against the erroneous information flow inside the model when it errs on more complex problems. This is among the reasons for our claim that our results will be useful to future mechanistic interpretability research on logical reasoning. This latter pursuit, however, deserves independent investigations, and is beyond the scope of the current work.
>
> **References**.
>
> 1. Honghua Zhang, Liunian Harold Li, Tao Meng, Kai-Wei Chang, and Guy Van Den Broeck. On the paradox of learning to reason from data. In IJCAI, 2023.
>
> 2. Xinyun Chen, Ryan A. Chi, Xuezhi Wang, and Denny Zhou. Premise order matters in reasoning with large language models, 2024. In ICML, 2024.
>
> 3. Matej Zecevic, Moritz Willig, Devendra Singh Dhami, and Kristian Kersting. Causal parrots: Large language models may talk causality but are not causal. In TMLR, 2023.
>
> 4. Kevin Ro Wang, Alexandre Variengien, Arthur Conmy, Buck Shlegeris, and Jacob Steinhardt. Interpretability in the wild: a circuit for indirect object identification in GPT-2 small. In ICLR, 2023.

---

> > ### Comment · Reviewer_qubH · 2024-11-28
> >
> > I am sorry, but then isn't entitling the paper "How Transformers Solve Propositional Logic Problems" slightly misleading, when actually considering a rather specific and simple type of propositional reasoning?
> >
> > It might still extends the state of the art, but it still looks quite far away from propositional reasoning.

---

> ### Author Response · Authors · 2024-11-28
>
> Thank you for your reply.
>
> First, we can make some minor changes to the title such as "How Transformers Solve Propositional Logic Problems: _Towards a Mechanistic Understanding_" in the camera-ready version of the work if it gets accepted.
>
> More importantly, while we understand your concerns on the generality of our analysis' setting, we still believe that our work makes a meaningful contribution towards understanding how LLMs solve (propositional) reasoning problems, and towards establishing more general and rigorous techniques of analyzing LLMs.
>
> In addition to what we had already discussed in our response above, including how our problem generalizes those studied in related circuit analysis works, and the resemblance of our problem to existing propositional-logic-based benchmark and analysis papers, we also provide some high-level reasons below.
>
> First, there typically is a trade-off between the _generality_ of the problem setting which the (mechanistic) analysis is performed in, and the _precision_ of the insights and conclusions reached. Obtaining precise insights on a subject often requires working in _simplified_ settings, and is especially so at the early phase of a subject of study when little results and techniques are established.
>
> Circuit analysis of LLMs, including our work, falls on the "precision" end of the spectrum, and is indeed in an early phase as a subject of study (especially on the sub-area of  logical reasoning), as discussed before. We choose to pursue as precise and in-depth an understanding of how LLMs solve propositional logic problems as possible, by localizing a _human-interpretable reasoning circuit_ inside the model (including Mistral-7B and Gemma-2-9B!) which satisfies necessity and sufficiency criteria. This analysis is done on problems which are (1) _complex_ enough such that the LLM _cannot_ skip any of the step in the reasoning chain of “_Question→Relevant rule(s) →Relevant fact(s) →Decision_”, and (2) _simple_ enough to facilitate _interpretation and causal intervention studies_.

---

> ### Author Response · Authors · 2024-12-03
>
> Dear Reviewer qubH,
>
> We hope we have addressed your concerns. As the deadline for the discussion period is quickly approaching, please let us know about anything in our work you would like further clarification on.

---

### Official Review · Reviewer_MVJS · 2024-11-01

**Soundness:** 3
**Presentation:** 2
**Contribution:** 3
**Rating:** 6
**Confidence:** 4

**Summary:**

The paper provides an analysis into how transformers may solve propositional logic problems which require some degree of reasoning and/or planning using the rules of logic to check whether a given statement is entailed using the provided facts. The authors investigate two different kinds of transformers. The first kind is a vanilla 3-layer 3-head transformer trained from scratch to solve such problems. The second is an investigation into the Mistral-7B LLM which has been trained on a large amount of data and seems to exhibit reasoning capabilities.

For the problem setup, the paper proposes a simple propositional logic problem with two different types. In the first type, two separate implication chains are connected together by a logical operator (assuming conjunction or disjunction since it is not clearly mentioned and these appear in lines 290-291). Thus, the antecedent is two chains of implications connected by an operator. The query is the consequent (a single proposition) and the query asks the truth value using the rules of inference. Similarly, the linear chain is simply a sequence of implications whose consequent is again a single symbol.

The QUERY token is the terminating node of the chain and in this paper seems to be a single proposition.

The paper trains a decoder-only transformer s.t. it attains 100% accuracy on the training set. It is unclear whether there is a test set since their is no mention of it. The LogOp chain is queried 80% of the time with the linear one being the remainder. Their evaluation consists of a string match where an answer is correct iff all output tokens exactly match the ground truth.

The authors then analyze the cosine similarity of the Query token for the LogOp and Linear chains and find that there exist special routing chains that existing depending upon whether a LogOp or Linear chain is being queried. One interesting ablation here is that for linear chains of length 3, the answer is already available at layer 2. The authors train a classifier using the output of this layer and show that it is able to achieve 97%  accuracy. The same ablation or data for the LogOp is not mentioned however and instead a separate analysis is performed.

The next analysis is that of Mistral 7B where the authors use causal mediation analysis and looks for differences in activation function values between the two type of queries and provide insights from these results as well. The authors provide some soft out-of-distribution (OOD) analyses here by swapping locations of linear rule for tokens.

**Strengths:**

S1. The paper makes a detailed insight into the reasoning capabilities of transformers.

S2. The paper is generally well-written and the topic is important for the broader AI community.

S3. The paper shows that transformers can (within reason) learn to reason since they appear to have planning circuits.

S4. The techniques employed are intuitive and convincing.

**Weaknesses:**

W1. The data model is quite simple and I am not sure if this analysis can provide a definite answer to a broader claim of whether LLMs can reason. While I appreciate the simplicity it is not clear why a transformer would not be able to perfectly learn such a simple data model. A single perception can learn the AND truth table so I am not sure why such a data model was selected.

W2. One complaint in the writing is that it is absolutely unclear whether there exists a train and test set. This forms a major part of the paper so I was surprised that it was not included. I am not sure if 100% accuracy is on the training set or the test set? Was there any OOD data?
What is the size of the training set? The data model is quite simple (no negations as well) and it would not take many samples to learn such a function I think.

W3. Ablation 3.2.2. is interesting but fails to provide accuracy for the LogOp chain. The hypothesis would make sense if that has low accuracy. Similarly, there is no data on reasoning lengths or reasoning chains of length 4, 2 etc. These ablations would provide more insights into the capabilities IMO. Do you have any data with a linear chain like $p_1 \rightarrow \ldots \rightarrow p_n$.

W4. Implications and Bi-conditions do not really exist in logic in the strict sense since they can be expressed using and, not, and or. Have the authors performed OOD tests that can replace one of the $a \rightarrow b$ statements with $\neg a \lor b$.

**Questions:**

Thank you for your detailed examination. This is quite interesting work. Please address my comments in the weaknesses. Im happy to discuss further. I hope the authors can resolve my concerns.

---

> ### Author Response · Authors · 2024-11-19
> **Response to Reviewer MVJS, Part 1**
>
> Thank you for your insightful comments and suggestions! Our response to your questions and comments is as follows.
>
> **Position and contribution of this work**. Before addressing your specific comments, we wish to clarify the goal of our work, and its position in the literature. First, as discussed in Section 1.1 of our paper, our paper belongs to the field of mechanistic interpretability, particularly the area of _circuit analysis_ of large language models. This area is new, and most existing studies have been performed on _highly simplified problems_ that grant accessibility to fine-grained analysis of information flow and processing inside the model, at the level of attention heads, MLPs, etc. [1-3]. For example, the pioneering circuit analysis work [1] studies the attention circuit in GPT2-small on a simple task of identifying the “missing variable” in a conjunctive sentence, where the prompt templates follow simple forms such as “When Mary _and_ John went to the store, John gave a drink to ”: the answer is “Mary”, the “missing variable” from the conjunctive sentence.
>
> The primary goal of our work is to _initiate_ the study of reasoning circuits in transformer-based language models, especially in pretrained LLMs. We wish to place special emphasis on our analysis of how Mistral-7B solves the reasoning problem, as it is, to the best of our knowledge, the first to characterize the critical components of the circuit employed by an LLM _in the wild_ for solving a nontrivial logical reasoning problem (involving distracting clauses) that requires a correct execution of the full reasoning path “Query→Relevant rule(s) →Relevant fact(s) →Decision'' fully in context. The reasoning problem, while simple, captures some of the core aspects of broader classes of reasoning problems in practice. Moreover, we believe that the techniques we developed for examining the necessity and sufficiency of the pretrained LLM's reasoning circuit is of concrete value to future research in _mechanistic interpretability_.
>
>
> **W1** & **W4**. “Data model is quite simple and I am not sure if this analysis can provide a definite answer to a broader claim of whether LLMs can reason”, “not clear why a transformer would not be able to perfectly learn such a simple data model”, “single perception can learn the AND truth table…”, “implications and Bi-conditions do not really exist in logic in the strict sense”, “no data on reasoning lengths or reasoning chains of length 4, 2 etc”.
>
> We wish to emphasize that, the theme of this paper is _not_ to answer whether LLMs can reason in general, but to determine whether there even exists a circuit (consisting of a small number of internal components of the LLM) which the LLM employs to solve a (simple) reasoning problem, and to frame the controlled experiments delicately such that we can localize and verify this circuit through causal intervention techniques. This means that, we need to choose problems which are (1) complex enough so that the problems are not solved through simple shortcuts (skipping steps in the “Query→Relevant rule(s) →Relevant fact(s) →Decision” chain we discussed above), and (2) simple enough to facilitate interpretation and intervention studies — this simplicity is especially vital for intervention studies of LLMs in the wild (Mistral-7B in our case).
>
> Furthermore, since the reasoning problem still requires the model to perform several actions that are essential to more complex reasoning problems, the analysis techniques and insights we developed can serve as the foundation for understanding not only how the LLM solves more complex reasoning problems, but also why it sometimes fails on those problems, by identifying anomalous information flow and processing in the LLM in those settings.
>
>
> **W2**. “unclear whether there exists a train and test set”, “Was there any OOD data? What is the size of the training set?”
>
> We apologize for the lack of clarity on this, we will add these details in the revision. The 100% accuracy we report is the _test_ accuracy. The small transformer (in Section 3) is trained on 2 million samples, and tested on 10k samples. We do not test them on OOD data.

---

> ### Author Response · Authors · 2024-11-19
> **Response to Reviewer MVJS, Part 2**
>
> **W3**. “Ablation 3.2.2. is interesting but fails to provide accuracy for the LogOp chain. The hypothesis would make sense if that has low accuracy”.
>
> Just to ensure that we are on the same page first, the purpose of the linear probing experiment in 3.2.2 (Evidence 2) is to verify that the answer for the linear chain is indeed linearly encoded in the layer-2 residual stream at the QUERY position, _when the linear chain is queried in the problem instance_. The reason we are performing this experiment is because of Evidence 1b (Lines 227-232): when the linear chain is queried, we find that the attention weights in the layer-3 attention heads are all almost 100% focused on the QUERY position. This naturally leads to the question of whether the layer-2 residual stream at this position already has the answer: we are just doing a confirmatory experiment in Evidence 2 that the answer is indeed present there, and is in fact linearly encoded.
>
> Now, following your suggestion and to add further contrasting evidence to our examinations, we trained a linear classifier with exactly the same task as before, except it is now asked to predict the answer of the LopOp chain. The training set size is also 5k, same as before. As expected, the classifier achieves a very low test accuracy of approximately **27%** when tested on 10k samples, and exhibits severe overfitting, with the training accuracy around 94%.
>
> **References**.
>
> 1. Kevin Ro Wang, Alexandre Variengien, Arthur Conmy, Buck Shlegeris, and Jacob Steinhardt.
> Interpretability in the wild: a circuit for indirect object identification in GPT-2 small. In
> The Eleventh International Conference on Learning Representations, 2023.
>
> 2. Michael Hanna, Ollie Liu, and Alexandre Variengien. How does gpt-2 compute greater-than? interpreting mathematical abilities in a pre-trained language model. In Proceedings of the 37th International Conference on Neural Information Processing Systems, 2023.
>
> 3. Nanda, Neel, Lawrence Chan, Tom Lieberum, Jess Smith, and Jacob Steinhardt (2023). “Progress measures for grokking via mechanistic interpretability”. In The Eleventh International Conference on Learning Representations, 2023.

---

> > ### Comment · Reviewer_MVJS · 2024-11-20
> >
> > Thank you for providing the necessary clarifications. My concerns on W2 and W3 have been resolved.
> >
> > I still have some reservations about the W1 and W4. Is there any reason why negations etc are not considered. I appreciate the analysis and fully understand the contributions being made. However, I am still reserved about the simplicity of the data model.
> >
> > In fact negations would be a perfect fit for your data model since $a \rightarrow b$ is nothing but $\neg a \lor b$. So you could consider a data model $(a \rightarrow b) \rightarrow c$ and analyze $(\neg a \lor b) \rightarrow c$ which resonates well with the LOGICOP in your paper.
> >
> > Please clarify.

---

> ### Author Response · Authors · 2024-11-20
> **Response to Reviewer MVJS's comment (Nov 19th)**
>
> Thank you for your speedy reply! There are several reasons for the choice of our reasoning problem.
>
> First, our reasoning problem resembles those studied in the recent propositional-logic-based benchmark and analysis papers such as [1-3]: in these works, negation is not considered, and all implications are written in the form “A implies B” (“If A then B”). In fact, [1] and [2] only contain propositional logic problems with definite clauses, and the only logical operator which possibly appears in a rule is conjunction; [2] restricts the problems even further by only allowing rules of the form $A \land B \to C$ or $A \to C$, with the answer always being True, and only require proper applications of modus ponens (they are mainly checking proof accuracy in their studies).
>
> Second, while “A implies B” and “Not A or B” are logically equivalent, the former is the much more frequently adopted way of conveying “implication” in normal (non-formal-logic)/internet writing, much of what general-purpose LLMs such as Mistral-7B are pretrained on and used for. To ensure relevance of our study to broader classes of reasoning problems faced by LLMs in practical settings while keeping the problem sufficiently simple, the more popularly written form “A implies B”/“If A then B" is chosen.
>
> Moreover, circuit analysis of pretrained LLMs in the wild requires significantly simplifying the problems of interest to facilitate precise interpretation and intervention studies. Consider for example the work [4], the most related circuit analysis work to ours, which focuses on problems with prompts of the form “When Mary and John went to the store, John gave a drink to ” [Answer is “Mary”], involving only a variable-relation statement. In contrast, we consider a nontrivial full-reasoning problem, demanding the LLM to properly process the variable relations, value assignments and query, and correctly ignore the distracting clauses.
>
> Now, we do consider negations an important subject in the study of how LLMs reason, however, due to its complexity, we believe that such a study demands a full project on its own, and is beyond the scope of a first reasoning circuit analysis. There are several directions that require careful investigations: (1) to understand how LLMs bind “NOT” to a proposition variable, and whether there even is a consistent circuit inside them in implementing such mechanisms, (2) to understand the circuits employed by an LLM in logically equivalent forms of “A implies B” and “Not A or B” and whether there is consistency, (3) to investigate whether LLMs can properly employ the contrapositive of existing rules to reason, and whether the "negation" sub-circuit in such inference settings are consistent with those in settings (1) and (2), etc.
>
> **References**.
>
> 1. Honghua Zhang, Liunian Harold Li, Tao Meng, Kai-Wei Chang, and Guy Van Den Broeck. On the paradox of learning to reason from data. In Proceedings of the Thirty-Second International Joint Conference on Artificial Intelligence, IJCAI, 2023.
>
> 2. Xinyun Chen, Ryan A. Chi, Xuezhi Wang, and Denny Zhou. Premise order matters in reasoning with large language models, 2024. In Forty-first International Conference on Machine Learning, 2024.
>
> 3. Matej Zecevic, Moritz Willig, Devendra Singh Dhami, and Kristian Kersting. Causal parrots: Large language models may talk causality but are not causal. Transactions on Machine Learning Research, 2023.
>
> 4. Kevin Ro Wang, Alexandre Variengien, Arthur Conmy, Buck Shlegeris, and Jacob Steinhardt. Interpretability in the wild: a circuit for indirect object identification in GPT-2 small. In The Eleventh International Conference on Learning Representations, 2023.

---

> > ### Comment · Reviewer_MVJS · 2024-11-20
> >
> > Thanks for providing the clarifications. In light of the relevant work in the area, I think that the paper makes a decent advance towards progressing our understanding of transformers w.r.t. formal logic.
> >
> > I'll increase my score by one point. I still think that the paper's impact would be further improved by either
> > * keeping the simplified data model but also considering first-order logic
> > * analyzing more than a single LLM and trying to contrast the differences.

---

> ### Author Response · Authors · 2024-11-28
> **Response to Review MVJS**
>
> > analyzing more than a single LLM and trying to contrast the differences.
>
> Thank you for your excellent suggestion!
>
> In the latest revision of our paper, **Appendix C.7** now includes a characterization of the reasoning circuit in Gemma-2-9B for solving the propositional problem we examined Mistral-7B on. **To our surprise, the reasoning circuit inside Gemma-2-9B bears strong resemblance to that in Mistral-7B**. In particular, we found that the circuit $C_{Gemma}$ (satisfying necessity and sufficiency criteria which the $C_{Mistral}$ circuit was also tested on) _also shares the four families of attention heads as Mistral-7B's based on their highly specialized attention patterns_:
> - $\text{Queried-Rule Locating Heads}$ $= \\{(19,11), (21,7), (22,5), (23,12)\\}$;
> - $\text{Queried-Rule Mover Heads} = \\{(20,7), (23,6), (24,15)\\}$;
> - $\text{Fact-Processing Heads} = \\{(24,5), (25,7), (26,0), (26,12)\\}$;
> - $\text{Decision Heads} = \\{(28,12), (30,9)\\}$.
>
> While it is too early to draw precise conclusions on how similar the two circuits in the two LLMs
> truly are (there currently is no rigorous metric for measuring circuit similarity at these circuits' level of complexity), the preliminary evidence suggests that the reasoning circuit we found in this work potentially has some degree of **universality** across LLMs.

---

> > ### Comment · Reviewer_MVJS · 2024-11-29
> >
> > Thanks for your additional experiments. I believe these findings significantly improve the papers impact. I've increased my score of  soundness of the paper.
> >
> > I've been following the remainder of the discussion here and I agree that this paper makes valuable contributions to the community and am recommending acceptance.

---

> > > ### Author Response · Authors · 2024-12-02
> > >
> > > Thank you for taking the time to review our paper updates and for your kind words of support. We appreciate your recommendation for acceptance!

---

### Official Review · Reviewer_H2ZQ · 2024-11-02

**Soundness:** 3
**Presentation:** 1
**Contribution:** 4
**Rating:** 6
**Confidence:** 3

**Summary:**

This paper presents a mechanistic analysis of how LLMs solve reasoning problems in propositional logic.
The authors first construct a problem template in which the correct reasoning must go through either a "LogOp chain" or a "Linear chain", where each problem instance randomly selects which chain to use.
Then, the authors analyze the behavior of a small GPT-2-like model on such problems and identify the behavior of each layer and each head.
Moreover, the authors use linear probing and activation patching to analyze the behavior of Mistral-7b on this dataset and find that there are four types of attention heads critical to logical decision-making.
This paper presents techniques with which researchers can better understand the reasoning mechanism of LLMs.

**Strengths:**

This paper presents an interesting experiment setup and thorough analysis.
I appreciate the pairing of small-model and large-model experiments: the small-model case lets one perform a more exhaustive analysis that then informs intuition for analyzing the large-model setting.
Of particular interest are the authors' findings for the four kinds of attention heads for Mistral-7b, and this is useful for better informing future directions.
Moreover, the techniques introduced for analyzing Mistral-7b would also be useful for other researchers.

**Weaknesses:**

The paper discusses many technical details with insufficient references to figures or mathematics.
This means that the reader can easily get lost.


For example, I had trouble following the technical details of Section 3.
I suspect this is because the paper assumes familiarity with the standard transformer architecture and uses terminology like "layer-2 embedding" and "layer-3 attention" without first introducing a reference figure or some mathematical definitions --- thus increasing the mental load on the reader.
For accessibility and clarity, I would greatly appreciate a figure with which key vocabularies are defined.
This is feasible because a 3-layer, 3-head model is quite small, and the Section 3 text makes heavy reference to many of these components.
Crucially, the 3-point description of the mechanism in Section 3.2 would benefit from such a reference figure.


Adding arrows on the lines would help readability in Figure 3.


Section 4.2's causal mediation analysis is hard to follow, and I think some examples would help the reader.

**Questions:**

What does Figure 2 show?

What was the motivating intuition that the first token (before CoT) might be important?

---

> ### Author Response · Authors · 2024-11-19
> **Response to Reviewer H2ZQ**
>
> Thank you for your valuable time and constructive feedback! The following is our response to your questions and comments.
>
> **W1**. “many technical details with insufficient references to figures or mathematics”, “technical details of Section 3”, “figure with which key vocabularies are defined”, "Adding arrows on the lines would help readability in Figure 3", “Section 4.2's causal mediation analysis is hard to follow, and I think some examples would help the reader”.
>
> Thank you for your detailed suggestions! We have incorporated them in our paper revision. In particular, we have updated Figure 3, and included a new Figure 7 in Appendix B.2.1 which illustrates the major components of the small transformer and the information flow inside it (we could not find enough space to add this large figure to Section 3 of the main text), and pointed out the more important locations/embeddings which we are primarily interested in.
>
> **Q1**. “What does Figure 2 show?”
>
> Briefly speaking, Figure 2 is there to show that the output embeddings from the layer-2 attention block at the QUERY token position exhibit disentanglement with respect to the chain type that is being queried, which we discuss in Evidence 1a.
>
> To understand Figure 2 in greater detail, we need to first recall its context. In our logical reasoning problem, there are two chains defining the rules: a logical-operator (LogOp) chain, and a linear chain; they do not share any common proposition variable. The QUERY token determines which chain the reasoner needs to provide an answer for. Intuitively speaking, this QUERY token is critical to the reasoner for determining the answer, as it determines the binary problem of which one of the two chains is being queried.
>
> Interestingly, the small transformer indeed makes significant use of the QUERY token, with Figure 2 illustrating one evidence for it. As we described in Lines 223 - 226, we generate this figure by randomly sampling 100 problem instances which query for the linear chain, and 100 problem instances querying for the LopOp chain (in Figure 2, indices 0 to 99 correspond to the former group, 100 to 199 correspond to the latter group). We then obtain the embedding from the layer-2 attention block at the QUERY token position, and compute the pairwise cosine similarity between all 200 samples. Figure 2 exhibits two interesting features: first, cross-group cosine similarity is negative (as shown in the top right and bottom left blocks), second, in-group cosine similarity is above 0 by a nontrivial margin. This indicates disentanglement in the layer-2 attention block’s embedding with respect to the chain type being queried.
>
>
> **Q2**. “What was the motivating intuition that the first token (before CoT) might be important?”
>
> We justify the importance of understanding how the LLM reasons without CoT on the first page of our paper, in Lines 45 to 54.
>
> Let us recap on the examples given there. “Rules: A or B implies C. D implies E. Facts: A is true. B is false. D is true. Question: what is the truth value of C?”
>
> At a high level, the intuition is that, since the first token requires the LLM to write down the correct fact in the minimal proof (“A is true” in our example), the LLM has to perform the following steps without CoT: first, it has to use the QUERY (in this example, C) to identify the queried rule, and then rely on the queried rule (with “premise” variables A and B) to identify the relevant facts, and finally make the decision that it should write down A as the first token, since “A is true” is the correct fact to invoke in the answer.
>
> Therefore, to write down the correct first answer token, the model has to fully process the context, and execute the chain of reasoning “QUERY→Relevant rule(s) →Relevant fact(s) →Decision” without CoT! Or to put it another way, the model has to form an “internal map” of the variable relations, value assignments and QUERY without CoT. We are interested in understanding the circuit inside the transformer for producing this “internal map” and how the components inside the circuit influence the LLM’s reasoning actions.

---

> > ### Comment · Reviewer_H2ZQ · 2024-11-21
> > **Response**
> >
> > Thank you for the reply. I will maintain my score.
> >
> > I believe that although the paper makes good contributions, its presentation deserves polishing so as to be worthy of an ICLR publication. For instance, while the new transformer definitions in Appendix B.2 are greatly appreciated, I feel that I can only understand the messy mathematics due to prior experience with these architectures. Moreover, I suspect that the current explanation of causal mediation analysis in Section 4.2 would be hard to follow for those not up-to-date with mechanistic interpretability tools. There are other such cases, but I cannot realistically expect the authors to address all of my many complaints nor rewrite entire sections in the short rebuttal period. It would also be a suboptimal use of the authors' time as there are other reviewers to deal with.
> >
> > In summary, interesting work, but please do more polishing!

---

> ### Author Response · Authors · 2024-11-25
> **Response to Reviewer H2ZQ's comments**
>
> Thank you for your positive assessment of our paper's contributions!
>
> We have uploaded a **new revision** which includes several new figures and improvements in writing to increase the clarity of the work further according to your suggestions:
> - We fully rewrote the mathematical definition of the small transformer architecture in **Appendix B.2**. This section also includes the new Figure 7 visualizing where the terms we frequently refer to in the main text are located in the model (including terms such as “layer-2 residual embedding”).
> - **Figure 8 in Appendix B.3** illustrates the _high-level strategy which the small transformer employs to solve the reasoning problem_, and includes summaries of the core observations for the problem solving strategies we discovered.
> - **Figure 10 and 11 in Appendix C.2** complement the explanations of our _causal mediation analyses_ in Section 4 of the main text, by visualizing the intervention experiments. In particular, Figure 10 illustrates the circuit discovery process, and Figure 11 illustrates the circuit verification process.
>
> Furthermore, we really appreciate your constructive feedback on improving this paper, so please feel free to share with us any further concerns you have about the paper! And of course, we will continue to polish our paper, and do our best to ensure the presentation quality of the camera-ready version of the work if it gets accepted.

---

### Author Response · Authors · 2024-11-19
**Manuscript updated**

Dear Reviewers and ACs,

We have uploaded a revision of the paper, with updated technical content, and incorporating reviewer suggestions to the best of our abilities within the time constraint.

We have highlighted the parts that underwent more significant changes in blue.

In the main text, Sections 1 and 4 underwent greater changes in general. Section 1 (Introduction) now provides more explanations on the position of this work in the literature and what our contributions are. Section 4 (Mistral-7B results) provides more technical details on the methodologies and their intuition, and a new subsection 4.3.2 which discusses _sufficiency_ evidence for the circuit we discovered, adding to the existing evidence of the circuit which was more necessity-based.

The Appendix underwent a significant update.

1. _Appendix B (small-transformer focused)_. In Appendix B.2, we provide cleaner definitions of the transformer architecture, and Figure 7 which visualizes the main locations inside the model which we frequently refer to in the main text. In Appendix B.3, we include a  new Figure 8 which illustrates the high-level strategy which the small transformer employs to solve the reasoning problem, and includes summaries of the core observations for the problem solving strategies we discovered.

2. _Appendix C (Mistral-7B focused)_. In Appendix C.2, we visualize the intervention experiments of our circuit discovery and verification processes of Mistral-7B, which complements the text descriptions of these experiments in Section 4.2 of the main text. In Appendix C.3, we discuss and visualize the attention statistics of the reasoning circuit's attention head families in greater detail. In Appendix C.4, we explain the finer details of the full-circuit verification experiments, and the general open problems with reasoning circuit verification or sufficiency tests. In Appendix C.5 and C.6, we provide fine-grained causal mediation experiments to examine how the attention head families function more precisely.

Authors of paper 13137

---

> ### Author Response · Authors · 2024-11-28
> **Latest revision with added results on Gemma-2-9B**
>
> In addition to the above updates, in the latest revision of our paper, **Appendix C.7** now includes a characterization of the reasoning circuit in **Gemma-2-9B** for solving the propositional problem we examined Mistral-7B on. **To our surprise, the reasoning circuit inside Gemma-2-9B bears strong resemblance to that in Mistral-7B**. In particular, we found that the circuit $C_{Gemma}$ (satisfying necessity and sufficiency criteria which the $C_{Mistral}$ circuit was also tested on) _also shares the four families of attention heads as Mistral-7B's based on their highly specialized attention patterns_:
> - $\text{Queried-Rule Locating Heads}$ $= \\{(19,11), (21,7), (22,5), (23,12)\\}$;
> - $\text{Queried-Rule Mover Heads} = \\{(20,7), (23,6), (24,15)\\}$;
> - $\text{Fact-Processing Heads} = \\{(24,5), (25,7), (26,0), (26,12)\\}$;
> - $\text{Decision Heads} = \\{(28,12), (30,9)\\}$.
>
> While it is too early to draw precise conclusions on how similar the two circuits in the two LLMs
> truly are (there currently is no rigorous metric for measuring circuit similarity at these circuits' level of complexity), the preliminary evidence suggests that the reasoning circuit we found in this work potentially has some degree of **universality** across LLMs.

---

### Meta-Review · Area_Chair_PVdW · 2024-12-18

**Metareview:**

The reviewers all agreed that there are some interesting parts of the paper -- specifically on Large LLMs, and the architectural implications of transformer. The reviewers also acknowledged that the paper goes beyond many papers for LLM reasoning.

However, the reviewers felt that problems considered are simplistic (and that the title is unncessarily overly broad, and the writing has to be improved to properly justify the setting.

**Additional Comments On Reviewer Discussion:**

The reviewers acknowledged the authors' rebuttal. It was also clear during the dicussions that a major rewrite is needed. Given that the rewrite might result in a different paper itself, a different round of reviewing is needed. Hence the reviewers all converged on a rejection with encouragement to submit after working through the issues.

I read the rebuttal closely and the message that the authors have sent me separately. But it is clear that some of the points raised by the reviewers (including tall claims) are something that need to be fixed before acceptance.

---

### Decision · Program_Chairs · 2025-01-22

Reject